# Longitudinal Evaluation of Large Language Models

## Abstract

Evaluations of large language models (LLMs) commonly report aggregate accuracy on static benchmarks, collapsing heterogeneous item behavior into a single score and obscuring how true performance evolves across model evolutions. This limitation is acute for rapidly evolving LLM families, where successive releases may progress unevenly or even regress. To mitigate this issue, we propose a dynamic Item Response Theory (IRT) framework for longitudinal evaluation that jointly calibrates benchmark items by difficulty and discrimination while modeling each LLM release with a time-varying latent ability. Applied to four multiple-choice benchmarks (JEE Math and three MMLU-Pro subsets) across GPT, DeepSeek, and Llama model families, with GPT used as the primary longitudinal case study, we observe non-monotone ability evolution and regimes where GPT-5.2 underperforms GPT-5 despite comparable accuracy, especially in MMLU-Pro Engineering. Item level diagnostics based on changes in model implied correctness probability show that regressions concentrate in the moderate difficulty and high discrimination region indicating strong evidence of reduced ability. These results demonstrate that dynamic IRT offers a task-sensitive, longitudinal lens for tracking progress and regressions beyond accuracy while separating true ability changes from the benchmark characteristics. Beyond model comparison, our framework offers a practical tool for benchmark designers and evaluators to identify which items carry meaningful evaluation signal, diagnose instability across model updates, and design more robust longitudinal evaluation protocols.

## 1 Introduction

Evaluating the capabilities of large language models (LLMs) is a central challenge in modern machine learning. Current evaluation approaches typically rely on aggregate accuracy over fixed benchmark item sets, implicitly treating all items as equally informative and equally difficult by assigning uniform weight to each item in the overall score. While accuracy is a convenient summary statistic, it conflates changes in model performance with heterogeneity in task difficulty and yields limited insight into where improvements (or regressions) occur. Moreover, accuracy is sensitive to the composition of the evaluation set: two models can have similar overall accuracy while differing substantially across easy versus hard items, or across different skill regimes. These limitations become more pronounced as benchmark suites grow in size and diversity, and as model families evolve rapidly across successive releases. A further challenge is that model updates are not guaranteed to be uniformly beneficial. Even when newer snapshots are expected to improve, regressions can occur due to changes in training data, post-training alignment, decoding behavior, or API-level updates Xu et al. (2025).

Recent survey papers emphasize that the evaluation of LLM remains an open and unresolved problem. Comprehensive reviews describe a rapidly expanding yet fragmented evaluation landscape encompassing diverse tasks, protocols, and scoring practices, and show that even seemingly minor variations in evaluation pipelines can lead to inconsistent or contradictory conclusions across studies Chang et al. (2024); Laskar et al. (2024). In parallel, a growing literature on benchmark design cautions that leaderboard style comparisons can be misleading, highlighting concerns such as data contamination, cultural and linguistic biases, and the absence of evaluation methodologies that meaningfully capture performance in dynamic or evolving environments Ni et al. (2025); Banerjee et al. (2024). More broadly, recent calls for a science of evaluating

foundation models advocate for structured, principled evaluation frameworks that promote reproducibility and yield actionable evidence beyond a single aggregate score Yuan et al. (2025).

These observations motivate evaluation approaches that explicitly acknowledge heterogeneity in benchmark items with respect to difficulty and informativeness, provide interpretable diagnostics that explain the sources of performance differences, and enable principled comparisons across snapshots as model families evolve over time. Item Response Theory (IRT) Baker & Kim (2004) provides a natural statistical framework for these goals by modeling the probability of a correct response through the interaction of a latent ability parameter and item characteristics such as difficulty and discrimination. More broadly, psychometric perspectives have encouraged a shift away from static benchmarking toward adaptive and data-efficient evaluation regimes, in which item parameters are explicitly estimated and test designs are optimized to maximize information Zhuang et al. (2023); Polo et al. (2024).

Recent works have begun to adapt IRT-based ideas for LLM evaluation, demonstrating the value of item-level modeling for difficulty-aware comparisons and evaluation diagnostics Pellert et al. (2023); Cheng et al. (2024); Ding et al. (2024); Park et al. (2024); Guinet et al. (2024); Li et al. (2025a;b); Truong et al. (2025); Polo et al. (2024). However, most existing approaches remain fundamentally static, assigning each model a single latent ability parameter estimated from a fixed snapshot of a benchmark. Assuming model ability to be independent over time is incorrect as model families evolve over time, the model ability of the snapshots can be expected to be strongly related through training lineage and incremental updates. Moreover, the model ability across releases may exhibit uneven performance changes across tasks and difficulty regimes. In such cases, fitting static IRT models at each time point would result in inflated standard errors of model ability estimates as temporal correlations in responses are not accounted for. Additionally, Fitting separate static IRT models at each time point yields abilities that are not directly comparable due to scale indeterminacy. Related concerns appear in recent work advocating for evolving evaluation protocols to address benchmark saturation and data leakage Jiang et al. (2024), further underscoring the need for evaluation frameworks that explicitly account for temporal effects—either through test design or through the measurement model.

In this paper, we introduce a dynamic 2-parameter logistic (2PL) IRT model in which each the time-varying LLM ability evolves according to an autoregressive dynamics. By combining item-level measurement with a longitudinal state-space prior, our framework estimates an LLM's effective ability trajectory over time and identifies the specific items that drive observed progressions or regressions in evaluation performance. We recast LLM evaluation as a problem of temporal inference over evolving model families, rather than point-in-time scoring.

Beyond model comparison, our framework provides item-level diagnostics that attribute observed progressions or regressions to specific regions of the difficulty–discrimination space, revealing when apparent performance changes reflect genuine shifts in ability rather than artifacts of benchmark composition. By explicitly modeling temporal dependence in latent ability, our approach separates measurement error from persistent trends and enables principled longitudinal comparisons across evolving model families. Broadly, dynamic IRT offers a unified measurement framework for tracking progress, diagnosing ability regressions, and designing evaluation protocols that remain informative as benchmarks and models evolve simultaneously.

## 1.1 Related Work

IRT has a long history in psychometrics and has increasingly been adopted within NLP as a tool for interpreting model performance through latent traits and item parameters. IRT has been used to improve the efficiency of human evaluation for chatbots, simultaneously assessing system quality and the informativeness of prompts, illustrating the broader role of IRT as both a scoring model and an auditing tool for evaluation items Sedoc & Ungar, 2020. More broadly, Lalor et al., 2024 provide a tutorial perspective on IRT in NLP, emphasizing its utility for dataset analysis, model comparison, and rethinking leaderboard-style evaluation. In the LLM context, Pellert et al., 2023 introduce "AI psychometrics", and Li et al., 2025b formalize IRT-based evaluation for LLMs, showing that calibrated item parameters and latent abilities yield more informative comparisons than raw accuracy.

A growing body of work aims to reduce evaluation cost and improve robustness by selecting informative items or constructing smaller but representative test sets. Zhuang et al., 2023 argue for a paradigm shift

from static benchmarks to adaptive testing for AI evaluation, explicitly estimating item characteristics and tailoring evaluation to the model. Polo et al., 2024 propose tinyBenchmarks, demonstrating that carefully curated subsets combined with IRT-based tools can reproduce benchmark-level estimates with far fewer examples. Relatedly, Cheng et al., 2024 and Li et al., 2025a develop adaptive testing protocols for efficient evaluation, while Guinet et al., 2024 use task-specific exam generation and item calibration to audit and refine evaluation sets.

Several works emphasize that benchmark items are not homogeneous and that difficulty stratification provides better diagnostics than aggregate accuracy. Park et al., 2024 study zero-shot difficulty estimation, and Ding et al., 2024 construct benchmarks with standardized difficulty labels to profile performance across regimes. Complementing these efforts, Jiang et al., 2024 propose generative evolving testing to address evaluation "chronoeffect" arising from benchmark saturation and leakage over time.

## 2   Methodology

We aim to examine LLM family evolution in a way that goes beyond point-in-time accuracy. To this end, we adopt a dynamic IRT framework where each model snapshot is treated as a test-taker, each benchmark question as a test item, and the observed correctness outcomes as noisy measurements of a latent capability. IRT models provide a natural starting point because they delineate item properties (difficulty and discrimination) from model capability (latent ability). Our key contribution is the introduction of a state-space formulation in which model ability evolves dynamically across releases, enabling longitudinal inference that disentangles persistent ability trends from transient fluctuations and measurement errors.

Consider $T$ model snapshots evaluated on an item bank of size $J$. Let $Y_{tj} \in \{0, 1\}$ denote whether snapshot $t$ answers item $j$ correctly. We define the observed index set $\Omega \subseteq \{1, \dots, T\} \times \{1, \dots, J\}$.

We use the two-parameter logistic (2PL) IRT model. Each item $j$ has difficulty $b_j \in \mathbb{R}$ and discrimination $a_j > 0$, and each snapshot $t$ has latent ability $\theta_t \in \mathbb{R}$,

$$Y_{tj} \mid \theta_t, a_j, b_j \sim \text{Bernoulli}(p_{tj}), p_{tj} = \sigma(\eta_{tj}), \eta_{tj} = a_j(\theta_t - b_j), \tag{1}$$

where $\sigma(x) = (1 + e^{-x})^{-1}$.

This representation clarifies why aggregate accuracy can be misleading: two models may match in overall accuracy yet differ substantially on high-difficulty items ($b_j$ large), or differ mainly on highly discriminative items ($a_j$ large). Calibrating $\{a_j, b_j\}$ therefore enables difficulty-aware comparisons and item-level attribution of performance changes.

Static IRT assigns a single ability score to each snapshot. In practice, however, we often evaluate families of models released over time, and we would like to quantify whether changes are smooth, abrupt, or regime-specific. We therefore model ability as a time-varying latent process.

We use an AR(1) state-space prior:

$$\theta_1 \sim N(0, 1), \theta_t = \phi \theta_{t-1} + \varepsilon_t, \varepsilon_t \sim N(0, \sigma^2), t = 2, \dots, T, \tag{2}$$

with $|\phi| < 1$ and $\sigma > 0$. This prior (i) shares statistical strength across releases and (ii) yields an interpretable trajectory: $\phi$ controls persistence and $\sigma$ controls volatility of changes.

Let $\theta = (\theta_1, \dots, \theta_T)$, $a = (a_1, \dots, a_J)$, and $b = (b_1, \dots, b_J)$. Using the shorthand $\eta_{tj} = a_j(\theta_t - b_j)$, the log-likelihood over $(t, j) \in \Omega$ is

$$\ell(\theta, a, b) = \sum_{(t,j) \in \Omega} \left[ y_{tj} \eta_{tj} - \log(1 + \exp(\eta_{tj})) \right]. \tag{3}$$

The AR(1) prior in equation 2 implies

$$\log p(\theta \mid \phi, \sigma)$$
$$= -\tfrac{1}{2} \theta_1^2 - \tfrac{1}{2\sigma^2} \sum_{t=2}^{T} (\theta_t - \phi \theta_{t-1})^2 - (T-1) \log \sigma + C, \tag{4}$$

where $C$ does not depend on the parameters. We estimate parameters via maximum a posteriori (MAP):

$$\widehat{\Theta} = \arg\max_{\Theta}\left\{\ell(\theta, a, b) + \log p(\theta \mid \phi, \sigma)\right\}, \tag{5}$$
$$\Theta = \{\theta_{1:T}, a_{1:J}, b_{1:J}, \phi, \sigma\}.$$

We optimize equation 5 using gradient-based methods. Let $p_{tj} = \sigma(\eta_{tj})$ and residuals $r_{tj} = y_{tj} - p_{tj}$. For compactness define the observed sets

$$\Omega_t := \{j : (t, j) \in \Omega\}, \qquad \Omega^j := \{t : (t, j) \in \Omega\}.$$

Then the log-likelihood gradients are

$$\frac{\partial \ell}{\partial \theta_t} = \sum_{j \in \Omega_t} r_{tj}\, a_j, \tag{6}$$

$$\frac{\partial \ell}{\partial b_j} = -\sum_{t \in \Omega^j} r_{tj}\, a_j, \tag{7}$$

$$\frac{\partial \ell}{\partial \log a_j} = \sum_{t \in \Omega^j} r_{tj}\, a_j (\theta_t - b_j), \tag{8}$$

where we optimize over $\log a_j$ to enforce $a_j > 0$.

For the AR(1) prior, let $u_t = \theta_t - \phi\theta_{t-1}$ for $t \geq 2$. Then

$$\frac{\partial}{\partial \theta_1} \log p(\theta \mid \phi, \sigma) = -\theta_1 + \tfrac{\phi}{\sigma^2} u_2, \tag{9}$$

$$\frac{\partial}{\partial \theta_t} \log p(\theta \mid \phi, \sigma) = -\tfrac{1}{\sigma^2} u_t + \tfrac{\phi}{\sigma^2} u_{t+1}, t = 2, \ldots, T-1, \tag{10}$$

$$\frac{\partial}{\partial \theta_T} \log p(\theta \mid \phi, \sigma) = -\tfrac{1}{\sigma^2} u_T, \tag{11}$$

$$\frac{\partial}{\partial \phi} \log p(\theta \mid \phi, \sigma) = \tfrac{1}{\sigma^2} \sum_{t=2}^{T} u_t\, \theta_{t-1}, \tag{12}$$

$$\frac{\partial}{\partial \log \sigma} \log p(\theta \mid \phi, \sigma) = -(T-1) + \tfrac{1}{\sigma^2} \sum_{t=2}^{T} u_t^2. \tag{13}$$

We maximize the MAP objective over the unconstrained parameterization

$$\{\theta_{1:T},\, b_{1:J},\, \log a_{1:J},\, \psi,\, \log \sigma\}, \qquad \phi = \tanh(\psi),$$

which guarantees $|\phi| < 1$. All sums are restricted to $\Omega$, so missing responses are handled without imputation.

2PL IRT admits shift/scale invariances: joint transformations of $(\theta_t, b_j, a_j)$ can leave equation 1 unchanged. We anchor the scale using the ability prior $\theta_1 \sim N(0, 1)$ together with the AR(1) dynamics in equation 2, and enforce $a_j > 0$ to preserve monotonicity of correctness probability in ability. These choices make $\theta_t$ comparable across releases as a latent capability measure. It is important to note that larger samples are required to obtain stable and reliable estimates of $(\theta_t, b_j, a_j)$ and their standard errors.

## 3 Experiments

Our experiments are designed to illustrate how a dynamic IRT framework complements standard accuracy reporting by (i) separating changes in measured performance from heterogeneity in item difficulty, (ii) enabling longitudinal comparison across sequential model releases, and (iii) exposing which regions of the item bank drive apparent gains and regressions.

Table 1: Benchmarks and fitted AR(1) parameters for dynamic ability across model families.

| Family | Benchmark | T | J | $\hat{\phi}$ | $\hat{\sigma}$ |
|---|---|---|---|---|---|
| GPT | JEE Math | 18 | 380 | 0.569 | 1.338 |
| GPT | MMLU-Pro (Engineering) | 18 | 969 | 0.547 | 0.335 |
| GPT | MMLU-Pro (Law) | 18 | 1101 | 0.603 | 0.444 |
| GPT | MMLU-Pro (Math) | 18 | 1351 | 1.000 | 3.705 |
| DeepSeek | JEE Math | 6 | 380 | 0.407 | 0.000 |
| DeepSeek | MMLU-Pro (Engineering) | 6 | 969 | 0.609 | 0.224 |
| DeepSeek | MMLU-Pro (Law) | 6 | 1101 | 0.188 | 0.048 |
| DeepSeek | MMLU-Pro (Math) | 6 | 1351 | -1.000 | 0.342 |
| Llama | JEE Math | 5 | 380 | 0.127 | 2.277 |
| Llama | MMLU-Pro (Engineering) | 5 | 969 | 0.996 | 0.980 |
| Llama | MMLU-Pro (Law) | 5 | 1101 | -1.000 | 2.044 |
| Llama | MMLU-Pro (Math) | 5 | 1351 | -0.987 | 2.600 |

We evaluate our method on four multiple-choice benchmarks: JEE Math Physicswallah AI Research (2025) and three subsets of MMLU-Pro Wang et al. (2024), namely Engineering, Law, and Math. For each benchmark, we form a binary response matrix $Y \in \{0,1\}^{T \times J}$ where $Y_{t,j} = 1$ indicates that the model snapshot at time index $t$ answers item $j$ correctly. Here $T$ denotes the number of evaluated model snapshots and $J$ denotes the number of items in the benchmark.

Table 1 summarizes the scale of each benchmark and the fitted dynamics parameters. Our primary longitudinal analysis uses 18 GPT-family snapshots, which provide the longest chronological sequence among the evaluated publicly available model families. To assess whether the proposed diagnostic framework is tied specifically to GPT-family models, we additionally evaluate shorter chronological sequences from DeepSeek and Llama. These additional families are treated as supporting evidence: they demonstrate that the same dynamic IRT pipeline can be applied across model ecosystems, while the detailed item level regression analysis is reserved for the longer GPT sequence.

We evaluate a sequence of model snapshots spanning multiple generations, from older GPT-3 series models to newer GPT-4/5 series models. To enable longitudinal inference, we assign each snapshot a time index based on chronological model order.

For each benchmark, we fit a dynamic 2PL IRT model with an autoregressive latent trajectory for ability. At each time index $t$, the model produces an ability estimate $\hat{\theta}_t$ which accounts for item difficulty and discrimination, rather than treating all items as equally informative. We report baseline accuracy per snapshot, the estimated latent ability trajectory, and the estimated item parameter distributions, to highlight the relationship between benchmark structure and observed performance.

Figure 1 shows the estimated latent ability trajectories $\hat{\theta}_t$ across releases for all four benchmarks. On JEE Math, the inferred ability remains relatively flat or stable for earlier snapshots and then exhibits a pronounced late-stage improvement, indicating that most separable gains or improvements on this benchmark are driven by the most recent generations. In contrast, MMLU-Pro (Engineering) and MMLU-Pro (Law) show a rapid early rise followed by a plateau, suggesting that later snapshots yield diminishing marginal gains on the calibrated item bank. MMLU-Pro (Math) exhibits the strongest fluctuations, consistent with the large fitted volatility $\hat{\sigma}$: the trajectory displays sharper rises and falls across releases, indicating that changes across snapshots are not uniformly beneficial even when aggregate accuracy may suggest incremental progress. These patterns highlight the value of a longitudinal latent scale. Ability trajectories can expose regime shifts and non-monotone evolution that is less visually apparent under raw accuracy aggregation.

To connect the latent scale to conventional reporting, Figure 2 plots the corresponding accuracy trajectories over time. While accuracy and inferred ability are broadly aligned, the two views are not equivalent. Accuracy assigns equal weight to all items, whereas the IRT-based ability estimate is inherently sensitive to item location on the 2PL response curve. Consequently, the dynamic IRT trajectories amplify improvements that

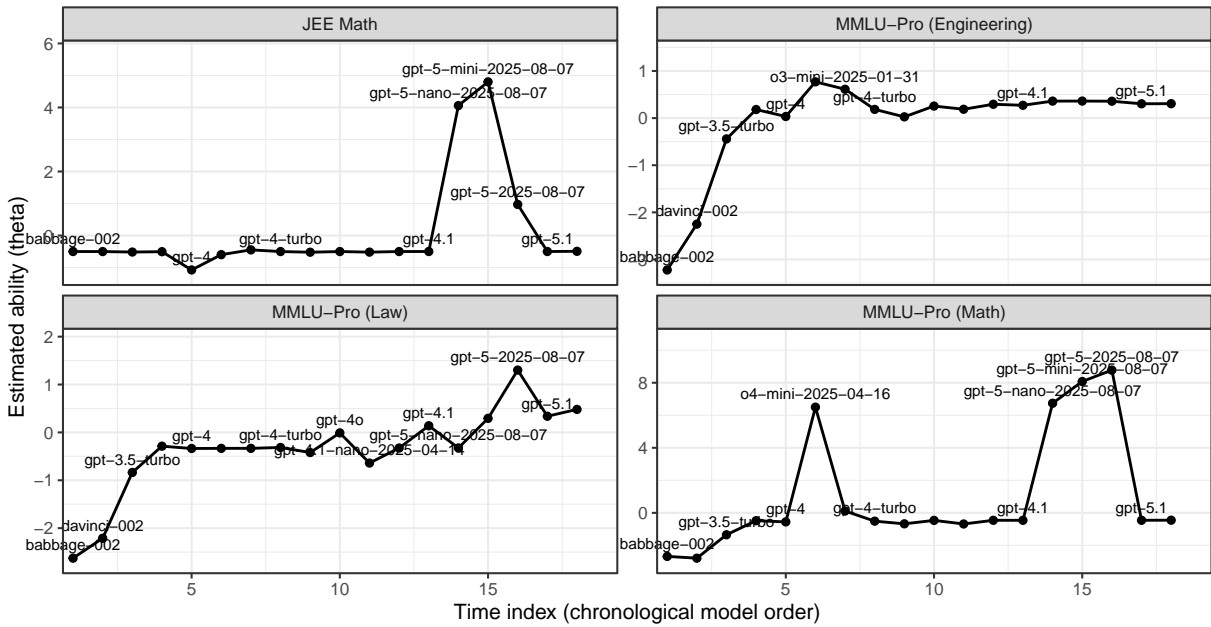

Figure 1: Estimated latent ability trajectories $\hat{\theta}_t$ across chronological model snapshots for each benchmark.

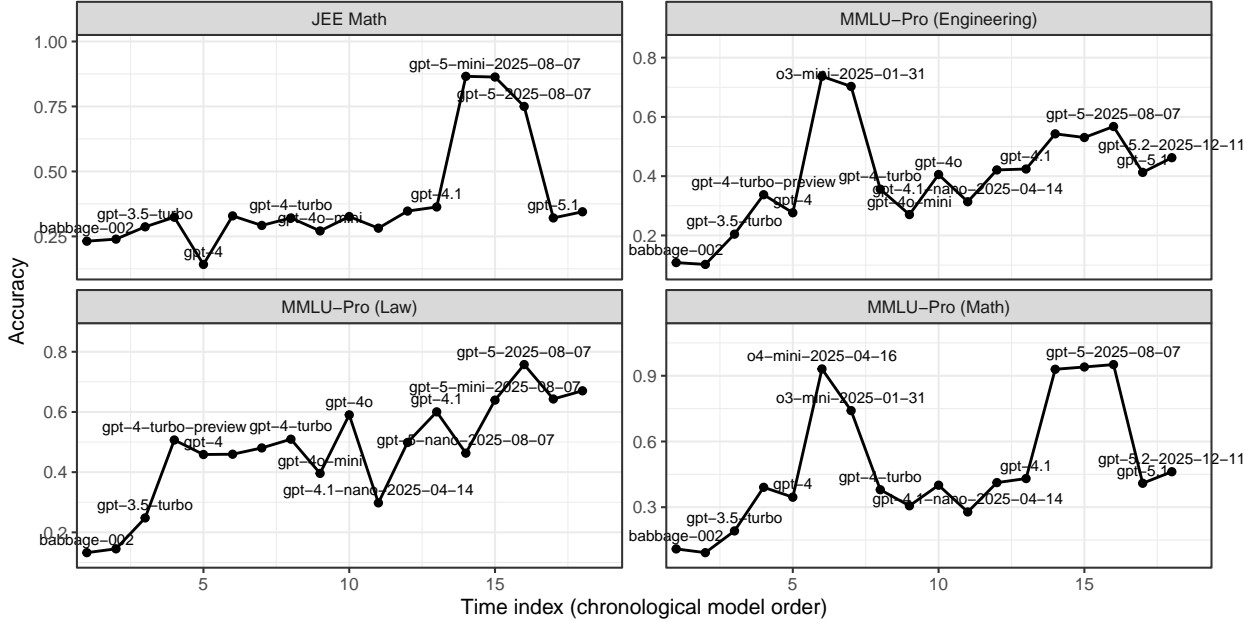

Figure 2: Accuracy trajectories across the same chronological model snapshots.

occur in the most informative regions of the benchmark items of intermediate difficulty with high discrimination, while downweighting changes driven primarily by very easy items, which contribute little marginal information once performance saturates. This distinction is particularly salient in the high accuracy regime, when a model already answers most easy items correctly, small changes on harder or more discriminative items may barely move accuracy, yet still induce meaningful shifts in inferred ability.

Finally, Figure 3 directly relates inferred ability to accuracy across snapshots. The relationship is mostly positive, serving as a sanity check that the inferred latent scale remains consistent with raw performance. However, the mapping is not perfectly linear, reflecting that accuracy depends on the mixture of item

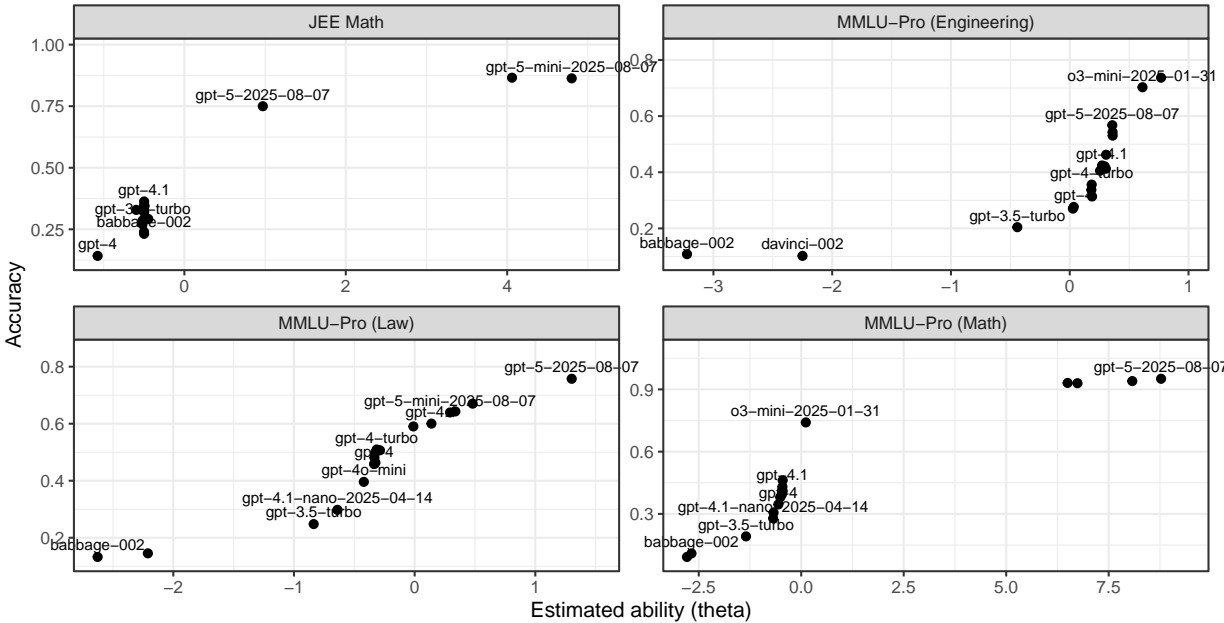

Figure 3: Scatter of accuracy versus inferred ability $\hat{\theta}$ across snapshots, faceted by benchmark. The Pearson correlation between accuracy and estimated ability (theta) are 0.93 (JEE Math), 0.75 (MMLU-Pro Engineering), 0.94 (MMLU-Pro Law), and 0.93 (MMLU-Pro Math).

difficulties and that IRT reweights responses through discrimination. In particular, two snapshots with similar accuracy can differ in inferred ability if one snapshot performs better on the most discriminative or intermediate-difficulty items, while the other gains mainly on easier or weakly informative items. This divergence underscores the central advantage of dynamic IRT: it complements accuracy by providing a more diagnostic, task-sensitive capability scale that supports longitudinal tracking and helps explain why apparent gains or regressions occur, rather than merely whether they occur. We emphasize that the GPT-family results constitute the main longitudinal case study because they provide the longest sequence of model snapshots. Results for DeepSeek and Llama are reported in Appendix B. These additional experiments serve as external-family checks: they show that the framework can be applied without modification to other model ecosystems, but we avoid making strong time-series claims from these shorter trajectories.

## 4 Dynamic IRT Diagnostics: Where does GPT-5.2 differ from GPT-5?

In this section we want to illustrate the use of dynamic IRT framework in identifying model ability progressions or regressions over evolutions, and the specific items from the benchmark that can reveal such patterns. While newer model versions are often expected to improve, our experiments show a net regression of GPT-5.2 relative to GPT-5 on several benchmarks. Such regressions under model/API updates have been reported previously and Ma et al., 2024 show that silent LLM API updates can degrade prompt performance, with many prompt–model combinations losing accuracy and previously-correct predictions flipping to incorrect. Echterhoff et al., 2024 similarly quantify substantial negative flips under model updates and propose compatibility-aware strategies to mitigate them. Motivated by this, we use item-level IRT diagnostics not only to attribute gains, but also to localize where regressions occur (easy vs. hard items, and low vs. high-discrimination items).

A central advantage of an IRT-based comparison is that it allows us to move beyond a single aggregate accuracy number and instead ask: on which kinds of items does the newer model improve, and where does it regress? Throughout, we compare GPT-5.2 against GPT-5 using item parameters $(a_i, b_i)$ and the estimated

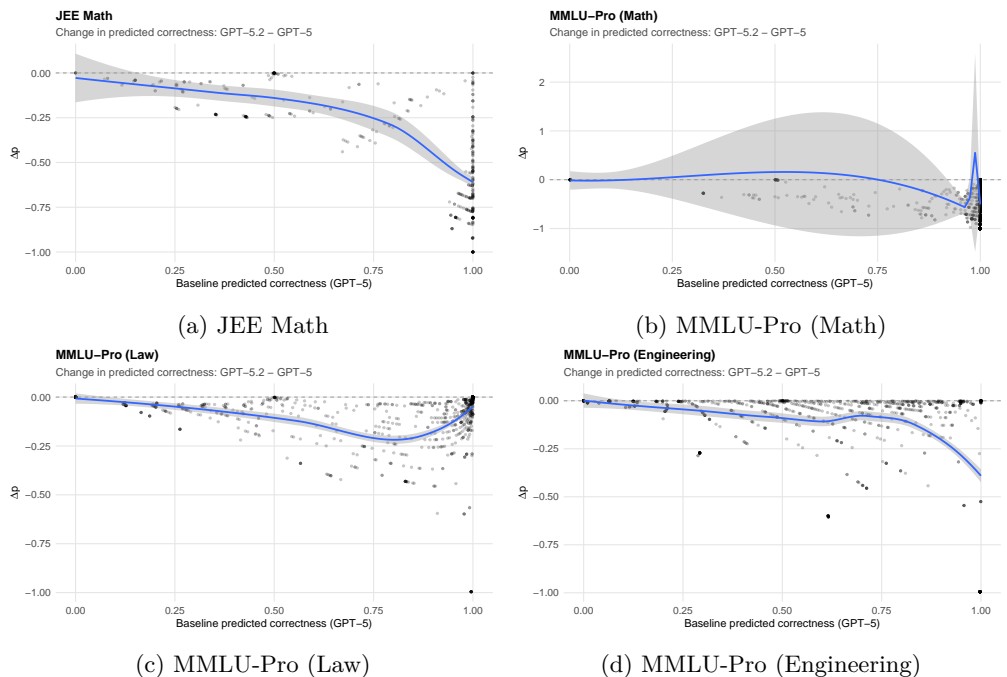

Figure 4: **GPT-5.2 vs GPT-5: improvement vs baseline correctness.** Each point is an item; the vertical axis is $\Delta p_i = p_i(\theta_{5.2}) - p_i(\theta_5)$, while the horizontal axis is the baseline probability $p_i^{(5)}$ under GPT-5. This diagnostic reveals whether improvements occur mainly on uncertain items (mid-range baseline), whether they saturate near the ceiling for easy items, and whether any regressions occur on items that GPT-5 previously solved reliably.

abilities $\theta_{5.2}, \theta_5$. For an item $i$, the model-implied probability of a correct response is

$$p_i(\theta) \;=\; \sigma(a_i(\theta - b_i)), \qquad \sigma(x) = \frac{1}{1 + e^{-x}}.$$

We define the item-level improvement signal as

$$\Delta p_i \;=\; p_i(\theta_{5.2}) - p_i(\theta_5),$$

so that $\Delta p_i > 0$ indicates an expected gain for GPT-5.2 on that item, while $\Delta p_i < 0$ indicates an expected regression.

### 4.1 Improvement vs baseline correctness

A natural first diagnostic is to relate $\Delta p_i$ to how likely the baseline model (GPT-5) already was to solve the item. Let

$$p_i^{(5)} := p_i(\theta_5)$$

be the GPT-5 baseline correctness probability on item $i$. Figure 4 plots $\Delta p_i$ against $p_i^{(5)}$.

This visualization is especially useful because the scale of possible improvement depends strongly on the baseline probability. If an item is already close to solved ($p_i^{(5)} \approx 1$), then $\Delta p_i$ is constrained by a ceiling effect. Conversely, if $p_i^{(5)}$ lies in an intermediate regime, then small changes in effective ability can translate into visibly larger shifts in predicted correctness, since the logistic curve is steepest away from the extremes.

Empirically, Figure 4 shows that the largest systematic deviations occur near the right end of the plot, i.e., for items that GPT-5 was already predicted to solve with high probability. In both JEE Math and MMLU-Pro (Engineering), the smoothed trend declines as $p_i^{(5)} \to 1$, indicating that GPT-5.2 exhibits a

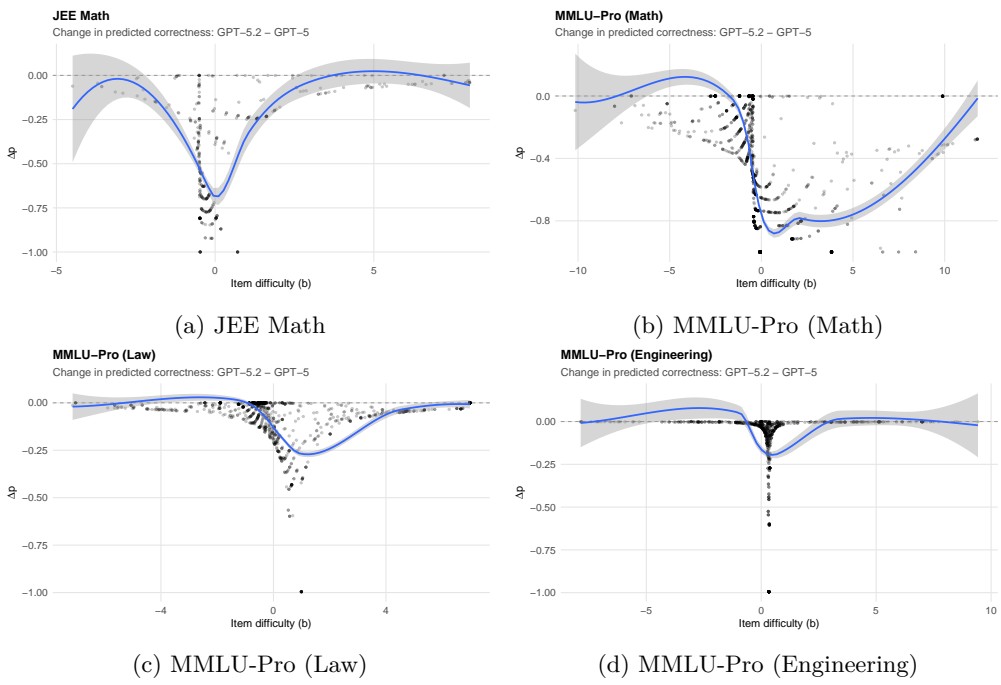

Figure 5: **GPT-5.2 vs GPT-5: improvement vs difficulty.** The horizontal axis is the estimated item difficulty $b_i$ (larger values indicate more difficult items). This diagnostic highlights whether the newer model's gains are concentrated in easy, medium, or hard regions of the test.

concentration of negative flips on items that were previously near-certain under GPT-5. This pattern is consistent with a regression on relatively easy questions: even small degradations in calibration, decoding, or instruction-following can cause occasional failures on items that a strong model normally answers reliably.

By contrast, for items where GPT-5 was uncertain ($p_i^{(5)}$ in the mid-range), the expected changes tend to be closer to 0 across benchmarks. This implies that GPT-5.2 is not uniformly worse across the entire spectrum, but instead that much of the regression signal is driven by items that were formerly in the "safe" region for GPT-5. Finally, the spread of points is widest in the mid-range of $p_i^{(5)}$ for several benchmarks, reflecting that this is where items are most sensitive to ability shifts and where heterogeneity in $(a_i, b_i)$ induces substantial item-level variation even for a fixed ability difference.

## 4.2 Improvement vs item difficulty

A second diagnostic is to examine how improvements vary with estimated difficulty $b_i$. Recall that in the 2PL model, $b_i$ acts as a location parameter, larger $b_i$ corresponds to more difficult items. Figure 5 plots $\Delta p_i$ against difficulty.

A useful interpretation comes directly from the 2PL link:

$$p_i(\theta) = \sigma(a_i(\theta - b_i)).$$

For fixed $(a_i, \theta_5, \theta_{5.2})$, $\Delta p_i$ is typically largest in magnitude when $b_i$ lies near the effective ability level, and becomes small for very easy or very hard items. Indeed, extremely easy items satisfy $b_i \ll \theta_5, \theta_{5.2}$, so both models have $p_i(\theta) \approx 1$ and $\Delta p_i \approx 0$. Similarly, extremely hard items satisfy $b_i \gg \theta_5, \theta_{5.2}$, so both models have $p_i(\theta) \approx 0$ and again $\Delta p_i \approx 0$. Thus, deviations concentrating in the intermediate difficulty region are particularly informative: they signal changes occurring near the benchmark's effective decision boundary.

Across benchmarks, Figure 5 shows that the regression signal is strongest at moderate difficulties. For JEE Math, the smoothed curve forms a pronounced negative valley around intermediate $b_i$, while tending back

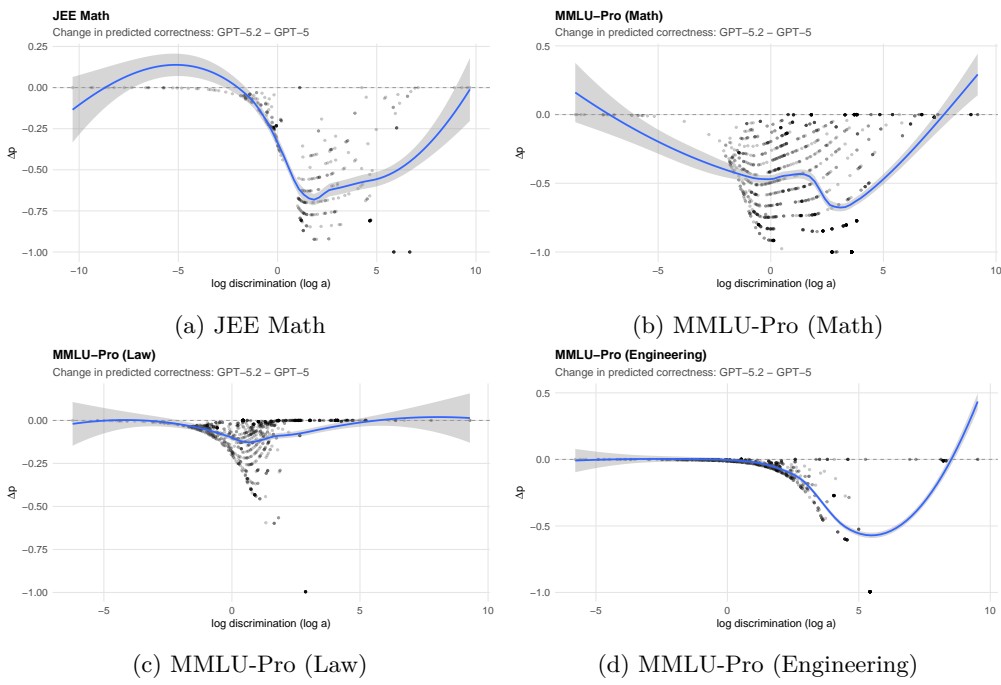

Figure 6: **GPT-5.2 vs GPT-5: improvement vs discrimination.** The horizontal axis is $\log(a_i)$, where $a_i$ is the item discrimination. High-discrimination items typically provide sharper separation between models; improvements concentrated here suggest meaningful ability gains.

toward 0 at the easiest end. This indicates that GPT-5.2 loses probability mass on the "core" region of the item bank that most cleanly separates adjacent ability levels. For MMLU-Pro (Math), the negative region is even broader and deeper, suggesting that the regression is not confined to a small slice of the benchmark, but persists across a wide band of medium-to-hard items.

In contrast, MMLU-Pro (Law) exhibits comparatively mild structure: the trend remains closer to 0, with only a shallow negative region, suggesting that GPT-5.2 is relatively stable on this subset and that its regressions, when present, are not strongly difficulty-selective. MMLU-Pro (Engineering) behaves more like JEE Math: the largest negative changes are concentrated around intermediate difficulties, with smaller deviations toward the extremes. Overall, the difficulty plots suggest that GPT-5.2's regressions are not primarily in the extreme hard tail, but rather on items that lie near the benchmark's central discriminative region, where the test provides the most evaluation signal (see *Tables 3 to 6* in Appendix for further details).

### 4.3 Improvement vs item discrimination

Finally, we relate improvements to item discrimination $a_i$, which controls how sharply correctness changes with ability. High-discrimination items (large $a_i$) are the ones that most strongly separate weaker from stronger models around the difficulty threshold. Figure 6 plots $\Delta p_i$ against $\log(a_i)$.

This diagnostic is important because discrimination determines how much a given ability shift matters. Even if two models differ only slightly in $\theta$, that difference can induce large probability changes on high-$a_i$ items, while producing almost no change on low-$a_i$ items. In this sense, regressions on highly discriminative questions are typically more consequential than regressions on weakly informative ones, since they correspond to items that most clearly distinguish capability levels in the benchmark.

In Figure 6, JEE Math and MMLU-Pro (Engineering) exhibit a clear negative region at moderate-to-high $\log(a_i)$, indicating that GPT-5.2's largest expected losses occur on items that are most diagnostic for ranking models. This is a particularly concerning failure mode from an evaluation standpoint: the regressions are

not limited to noisy, low-information items, but appear exactly in the region where the benchmark is most sensitive to real capability differences.

MMLU-Pro (Math) also shows strong negative trends across a wide range of discrimination values, reinforcing the conclusion from the difficulty plot that the regression is broad rather than narrowly localized. In contrast, MMLU-Pro (Law) again appears more stable, with changes closer to 0 across discrimination levels, suggesting that this subset contains fewer high-signal failures or that GPT-5.2's differences relative to GPT-5 are weaker on the most discriminative Law items.

Taken together, the discrimination plots help explain why an IRT-based comparison can differ from raw accuracy: a small number of failures on highly discriminative items can meaningfully reduce inferred ability, even when aggregate accuracy changes only modestly.

## 5 Conclusion

We presented a dynamic Item Response Theory (IRT) framework for longitudinal evaluation of evolving LLM families. By jointly calibrating item difficulty and discrimination and modeling ability as a time-varying latent trajectory, the method supports difficulty comparison across releases and item-level attribution of gains or regressions. Across multiple benchmarks, we find that ability evolution can be non-monotone, and that item-level diagnostics reveal changes that are obscured by aggregate accuracy. We find that negative changes concentrate on moderately difficult, highly discriminative items, the core evaluative region of the benchmark, explaining why small accuracy differences can mask substantial ability losses. This highlights a key advantage of our framework: it distinguishes genuine regressions from shifts driven by easy or weakly informative items. Dynamic IRT also supports principled monitoring of model updates, detection of silent regressions, and evaluation of benchmark informativeness as models evolve. Dynamic IRT framework also helps identify better items that can signify specific areas where progression or regressions occur. This method can also prune items from benchmark assessments thereby reducing computation burden while maintaining equivalent information about model ability.

Our current formulation focuses on binary multiple-choice outcomes with a unidimensional 2PL model and AR(1) dynamics. However, language models can perform tasks which are more complex and therefore extensions to partial-credit settings and multidimensional traits are promising directions of investigations. Overall, the dynamic IRT is a unifying measurement framework for longitudinal LLM evaluation that aligns with accuracy at a coarse level while offering deeper diagnostic insight into how, where, and why model ability trajectory changes.

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

# Appendix

## A  Experiment design

We evaluate a chronological sequence of OpenAI model snapshots (Table 2) using the OpenAI API.

Table 2: Model snapshots used in the analysis (chronological order).

| # | Model ID |
|---|----------|
| 1 | babbage-002 |
| 2 | davinci-002 |
| 3 | gpt-3.5-turbo |
| 4 | gpt-4 |
| 5 | gpt-4-turbo-preview |
| 6 | gpt-4-turbo |
| 7 | gpt-4o |
| 8 | gpt-4o-mini |
| 9 | gpt-4.1 |
| 10 | o3-mini-2025-01-31 |
| 11 | gpt-4.1-mini-2025-04-14 |
| 12 | gpt-4.1-nano-2025-04-14 |
| 13 | o4-mini-2025-04-16 |
| 14 | gpt-5-2025-08-07 |
| 15 | gpt-5-mini-2025-08-07 |
| 16 | gpt-5-nano-2025-08-07 |
| 17 | gpt-5.1 |
| 18 | gpt-5.2-2025-12-11 |

We also evaluate chronological sequences of DeepSeek and Llama model snapshots (Tables 3 and 4) using the OpenRouter API.

Table 3: DeepSeek model snapshots used in the analysis (chronological order).

| # | Model ID |
|---|----------|
| 1 | deepseek-chat |
| 2 | deepseek-chat-v3-0324 |
| 3 | deepseek-chat-v3.1 |
| 4 | deepseek-v3.1-terminus |
| 5 | deepseek-v3.2-exp |
| 6 | deepseek-v3.2 |

Table 4: Llama model snapshots used in the analysis (chronological order).

| # | Model ID |
|---|----------|
| 1 | llama-3-70b-instruct |
| 2 | llama-3.1-70b-instruct |
| 3 | llama-3.2-11b-vision-instruct |
| 4 | llama-3.3-70b-instruct |
| 5 | llama-4-maverick |

We use two multiple-choice benchmarks, MMLU-Pro and JEE Math. For MMLU-Pro, we focus on the subject subsets `Engineering`, `Law`, and `Math`.

For each item, we construct a single-turn multiple-choice prompt containing the question text and lettered answer options. To ensure consistent parsing and to discourage verbose completions, we include a strict output constraint requiring the model to return *only* the final option letter.

Let $\mathcal{O}$ denote the set of available option letters present in the benchmark item. The user prompt has the form:

```
Question: <question text>
Options:
<choices_formatted>
Please choose one option.
Constraint: Output ONLY the single letter (<allowed letters>).
Do not provide any explanation or text.
Answer:
```

The constraint string is adapted dynamically based on the number of answer choices:

- For 4 options: `A, B, C, or D`;

- For 5–6 options: an explicit list (e.g., `A, B, C, D, or E`);

- For up to 10 options: a range constraint (e.g., `A through J`).

For chat-based models, we additionally include a `system` message emphasizing that the task is multiple-choice and that the response must be a single option letter.

Each model snapshot is queried via API calls, with a small retry mechanism to handle transient failures and occasional rate limits. Two endpoints are used depending on model family. Legacy completion models (e.g., `davinci-002`, `babbage-002`) are queried using the Completions endpoint. Modern chat models are queried using the Chat Completions endpoint with a two-message structure (`system` and `user`).

To minimize stochasticity, we use `temperature = 0` whenever supported. For model families that do not accept explicit temperature control, the request is issued without this field and the API defaults are used. Since the target

output is a single option letter, the completion length is kept small (e.g., via a small `max_tokens` budget for completion-style models).

Raw model outputs are converted into binary correctness labels by extracting a predicted option letter and comparing it to the benchmark answer key. Specifically, we extract the first valid standalone letter in `A-J` from the model output using a regular expression. The extracted letter is treated as the model's predicted answer for item $i$ under model snapshot $t$.

An item is marked correct if the extracted letter matches the ground-truth key.

Figure 7 plots the distributions of estimated item difficulty and discrimination. Across all four benchmarks difficulty spans a wide range. Discrimination exhibits strong skew, indicating that a relatively small subset of items provides most of the separation signal between snapshots. This offers a direct explanation for why accuracy can be insensitive to certain changes: errors on low-discrimination items behave like noisy perturbations, whereas errors on high-discrimination items strongly influence inferred ability.

## B    DeepSeek and Llama trajectory diagnostics

Figure 8 reports the accuracy trajectories for the DeepSeek and Llama model families. Figure 9 compares accuracy with inferred latent ability for the same model families. As the DeepSeek and Llama sequences are shorter than the GPT sequence, we interpret these results primarily as portability checks rather than as the basis for detailed claims about family-specific evolution. For DeepSeek, the observed accuracy trajectories vary over relatively narrow ranges across benchmarks, and the corresponding ability–accuracy plots show that the inferred latent scale remains broadly aligned with raw accuracy. For Llama, the trajectories exhibit larger changes across snapshots, illustrating that chronological model order need not correspond to monotone benchmark improvement.

## C    Item-level comparisons between GPT-5 and GPT-5.2

This section provides representative items on which GPT-5.2 is at par with GPT-5 and items on which it underperforms relative to GPT-5. For each

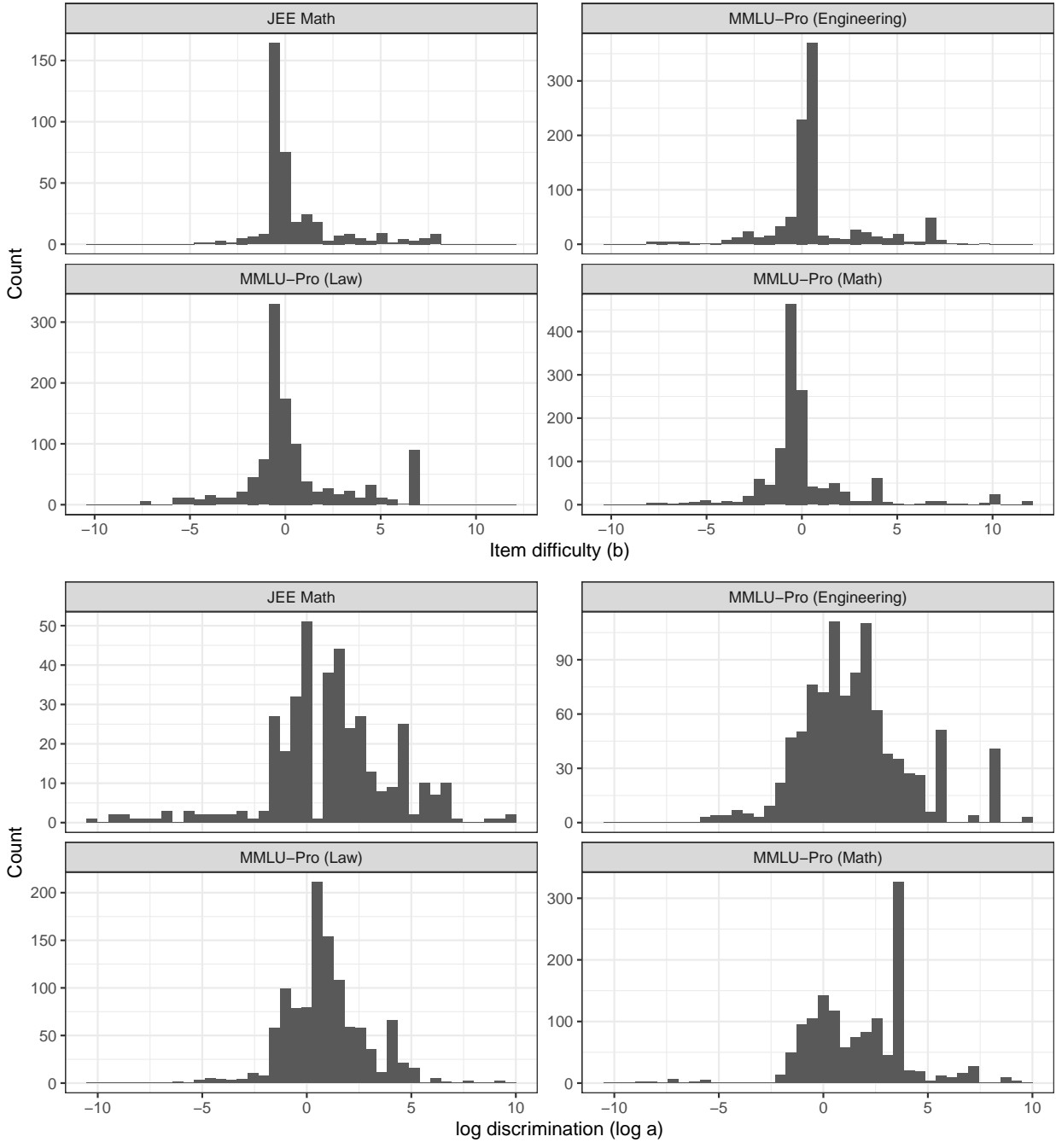

Figure 7: Top: distribution of estimated item difficulty $b_j$. Bottom: distribution of $\log(a_j)$, where $a_j$ is item discrimination.

item we report the model-implied improvement signal $\Delta p_i = p_i(\theta_{5.2}) - p_i(\theta_5)$ together with the observed binary outcomes from the benchmark runs.

### C.1 JEE Math selected items

**Item V350**  $b = $ -0.510,$a = $ 1194.615,$\Delta p = $ -0.000
**Observed correctness:** GPT-5 = 1, GPT-5.2 = 1

Table 5: Selected diagnostic items for JEE Math

| benchmark | item_id | difficulty | discrimination |
|---|---|---|---|
| JEE Math | V350 | -0.510 | 1194.615 |
| JEE Math | V280 | -0.500 | 16241.319 |
| JEE Math | V191 | -0.500 | 14356.160 |
| JEE Math | V342 | -0.500 | 7420.889 |
| JEE Math | V158 | -0.499 | 10794.618 |
| JEE Math | V113 | -0.470 | 351.273 |
| JEE Math | V147 | -0.470 | 351.273 |
| JEE Math | V323 | -0.470 | 351.273 |
| JEE Math | V327 | -0.470 | 351.273 |
| JEE Math | V329 | -0.470 | 351.273 |
| JEE Math | V333 | -0.470 | 351.273 |
| JEE Math | V218 | -0.209 | 0.000 |
| JEE Math | V243 | 0.689 | 0.000 |
| JEE Math | V118 | 0.708 | 784.018 |
| JEE Math | V122 | 0.708 | 784.018 |
| JEE Math | V129 | 0.708 | 784.018 |
| JEE Math | V379 | 0.708 | 784.018 |
| JEE Math | V43 | 0.708 | 784.018 |
| JEE Math | V51 | 0.708 | 784.018 |
| JEE Math | V160 | 2.440 | 0.000 |
| JEE Math | V194 | 4.702 | 69.800 |
| JEE Math | V343 | 7.214 | 3.053 |
| JEE Math | V358 | 7.214 | 3.053 |
| JEE Math | V39 | 7.214 | 3.053 |

Table 6: Selected items for MMLU-Pro (Engineering)

| benchmark | item_id | difficulty | discrimination |
|---|---|---|---|
| MMLU-Pro (Engineering) | V219 | -3.044 | 44.856 |
| MMLU-Pro (Engineering) | V231 | -3.044 | 44.856 |
| MMLU-Pro (Engineering) | V255 | -3.044 | 44.856 |
| MMLU-Pro (Engineering) | V335 | -3.044 | 44.856 |
| MMLU-Pro (Engineering) | V336 | -3.044 | 44.856 |
| MMLU-Pro (Engineering) | V229 | 0.030 | 1304.510 |
| MMLU-Pro (Engineering) | V237 | 0.030 | 1304.510 |
| MMLU-Pro (Engineering) | V251 | 0.183 | 3376.644 |
| MMLU-Pro (Engineering) | V57 | 0.183 | 3376.644 |
| MMLU-Pro (Engineering) | V216 | 0.186 | 3351.787 |
| MMLU-Pro (Engineering) | V12 | 0.332 | 226.860 |
| MMLU-Pro (Engineering) | V18 | 0.332 | 226.860 |
| MMLU-Pro (Engineering) | V25 | 0.332 | 226.860 |
| MMLU-Pro (Engineering) | V37 | 0.332 | 226.860 |
| MMLU-Pro (Engineering) | V43 | 0.332 | 226.860 |
| MMLU-Pro (Engineering) | V45 | 0.332 | 226.860 |
| MMLU-Pro (Engineering) | V49 | 0.332 | 226.860 |
| MMLU-Pro (Engineering) | V54 | 0.332 | 226.860 |
| MMLU-Pro (Engineering) | V77 | 0.332 | 226.860 |
| MMLU-Pro (Engineering) | V81 | 0.332 | 226.860 |
| MMLU-Pro (Engineering) | V84 | 0.332 | 226.860 |
| MMLU-Pro (Engineering) | V93 | 0.332 | 226.860 |
| MMLU-Pro (Engineering) | V455 | 0.361 | 13654.942 |
| MMLU-Pro (Engineering) | V566 | 0.361 | 13654.942 |

Table 7: Selected diagnostic items for MMLU-Pro (Law)

| benchmark | item_id | difficulty | discrimination |
| --- | --- | --- | --- |
| MMLU-Pro (Law) | V46 | -2.436 | 30.615 |
| MMLU-Pro (Law) | V82 | -2.436 | 30.615 |
| MMLU-Pro (Law) | V48 | -1.889 | 21.073 |
| MMLU-Pro (Law) | V67 | -1.889 | 21.073 |
| MMLU-Pro (Law) | V69 | -1.889 | 21.073 |
| MMLU-Pro (Law) | V76 | -1.889 | 21.073 |
| MMLU-Pro (Law) | V92 | -0.739 | 57.469 |
| MMLU-Pro (Law) | V23 | -0.533 | 61.504 |
| MMLU-Pro (Law) | V36 | -0.533 | 61.504 |
| MMLU-Pro (Law) | V105 | -0.335 | 2397.262 |
| MMLU-Pro (Law) | V31 | -0.328 | 145.265 |
| MMLU-Pro (Law) | V94 | -0.305 | 59.831 |
| MMLU-Pro (Law) | V355 | 0.574 | 5.217 |
| MMLU-Pro (Law) | V296 | 0.994 | 17.720 |
| MMLU-Pro (Law) | V370 | 0.994 | 17.720 |
| MMLU-Pro (Law) | V376 | 0.994 | 17.720 |
| MMLU-Pro (Law) | V39 | 0.994 | 17.720 |
| MMLU-Pro (Law) | V438 | 0.994 | 17.720 |
| MMLU-Pro (Law) | V582 | 0.994 | 17.720 |
| MMLU-Pro (Law) | V85 | 0.994 | 17.720 |
| MMLU-Pro (Law) | V878 | 0.994 | 17.720 |
| MMLU-Pro (Law) | V933 | 0.994 | 17.720 |
| MMLU-Pro (Law) | V947 | 0.994 | 17.720 |
| MMLU-Pro (Law) | V968 | 0.994 | 17.720 |

Table 8: Selected diagnostic items for MMLU-Pro (Math)

| benchmark | item_id | difficulty | discrimination |
|---|---|---|---|
| MMLU-Pro (Math) | V27 | -2.732 | 85.589 |
| MMLU-Pro (Math) | V101 | -2.516 | 40.915 |
| MMLU-Pro (Math) | V116 | -2.516 | 40.915 |
| MMLU-Pro (Math) | V125 | -2.516 | 40.915 |
| MMLU-Pro (Math) | V137 | -2.516 | 40.915 |
| MMLU-Pro (Math) | V28 | -2.516 | 40.915 |
| MMLU-Pro (Math) | V33 | -2.516 | 40.915 |
| MMLU-Pro (Math) | V4 | -2.516 | 40.915 |
| MMLU-Pro (Math) | V9 | -2.516 | 40.915 |
| MMLU-Pro (Math) | V105 | -0.457 | 7226.276 |
| MMLU-Pro (Math) | V109 | -0.457 | 7226.276 |
| MMLU-Pro (Math) | V36 | -0.457 | 7226.276 |
| MMLU-Pro (Math) | V1017 | 3.828 | 15.157 |
| MMLU-Pro (Math) | V1025 | 3.828 | 15.157 |
| MMLU-Pro (Math) | V1032 | 3.828 | 15.157 |
| MMLU-Pro (Math) | V936 | 3.828 | 15.157 |
| MMLU-Pro (Math) | V972 | 3.828 | 15.157 |
| MMLU-Pro (Math) | V976 | 3.828 | 15.157 |
| MMLU-Pro (Math) | V980 | 3.828 | 15.157 |
| MMLU-Pro (Math) | V981 | 3.828 | 15.157 |
| MMLU-Pro (Math) | V985 | 3.828 | 15.157 |
| MMLU-Pro (Math) | V996 | 3.828 | 15.157 |
| MMLU-Pro (Math) | V997 | 3.828 | 15.157 |
| MMLU-Pro (Math) | V66 | 6.619 | 36.876 |

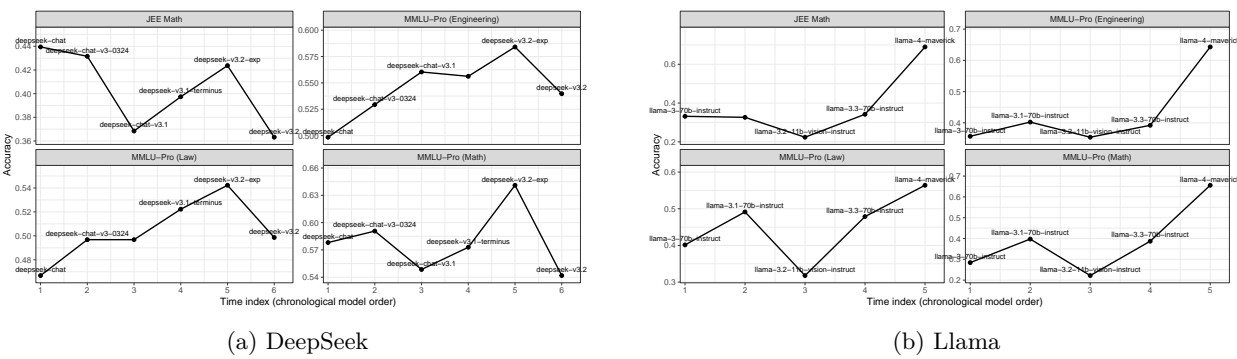

(a) DeepSeek

(b) Llama

Figure 8: Accuracy trajectories for the DeepSeek and Llama model families across chronological model snapshots.

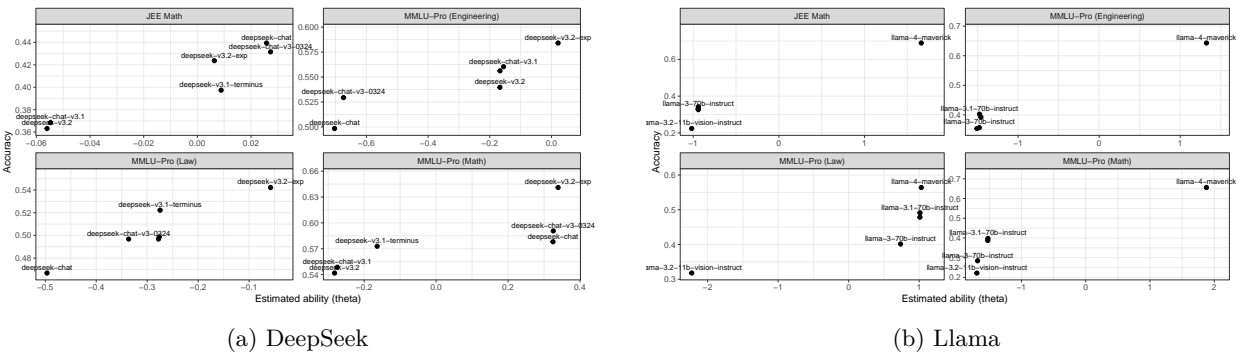

(a) DeepSeek

(b) Llama

Figure 9: Accuracy versus inferred ability $\hat{\theta}$ for the DeepSeek and Llama model families.

**Question:** Let $C_1$ be the circle in the third quadrant of radius 3, that touches both coordinate axes. Let $C_2$ be the circle with centre $(1, 3)$ that touches $C_1$ externally at the point $(\alpha, \beta)$. If $(\beta - \alpha)^2 = \frac{m}{n}$, $\gcd(m, n) = 1$, then $m + n$ is equal to:

**Choices:**
A. 9 B. 13 C. 22 D. 31

**Correct answer:** C

**Item V280** $b = -0.500, a = 16241.319, \Delta p = 0.000$
**Observed correctness:** GPT-5 = 1, GPT-5.2 = 1

**Question:** Let $f(x) = \begin{cases} (1 + ax)^{1/x}, & x < 0, \\ 1 + b, & x = 0, \\ \dfrac{\sqrt{x + 4} - 2}{\sqrt[3]{x + c} - 2}, & x > 0 \end{cases}$ be continuous at $x = 0$. Then $e^a bc$ is equal to:

**Choices:**
A. 64 B. 72 C. 48 D. 36

**Correct answer:** C

**Item V191** $b = -0.500, a = 14356.160, \Delta p = 0.000$
**Observed correctness:** GPT-5 = 1, GPT-5.2 = 1

**Question:** The system of equations $x + y + z = 6$, $x + 2y + 5z = 9$, $x + 5y + \lambda z = \mu$ has no solution if:

**Choices:**

A. $\lambda = 15, \mu \neq 17$ B. $\lambda \neq 17, \mu \neq 18$ C. $\lambda = 17, \mu \neq 18$ D. $\lambda = 17, \mu = 18$

**Correct answer:** C

**Item V342**   $b = \text{-}0.500, a = 7420.889, \Delta p = 0.000$
**Observed correctness:** GPT-5 = 1, GPT-5.2 = 1

**Question:** Evaluate the limit: $\lim_{x \to \infty} \frac{\tan\left(5x^{1/3}\right) \log_e(1+3x^2)}{\left(\tan^{-1}(3\sqrt{x})\right)^2 \left(e^{5x^{4/3}} - 1\right)}$ is equal to:

**Choices:**
A. 1/15 B. 1 C. 1/3 D. 5/3

**Correct answer:** C

**Item V158**   $b = \text{-}0.499, a = 10794.618, \Delta p = 0.000$
**Observed correctness:** GPT-5 = 1, GPT-5.2 = 1

**Question:** Let $(2,3)$ be the largest open interval in which the function $f(x) = 2\log_e(x-2) - x^2 + ax + 1$ is strictly increasing and $(b,c)$ be the largest open interval in which the function $g(x) = (x-1)^3(x+2-a)^2$ is strictly decreasing. Then $100(a+b-c)$ is equal to:

**Choices:**
A. 420 B. 360 C. 160 D. 280

**Correct answer:** B

**Item V113**   $b = \text{-}0.470, a = 351.273, \Delta p = \text{-}1.000$
**Observed correctness:** GPT-5 = 1, GPT-5.2 = 0

**Question:** The remainder, when $7^{103}$ is divided by 23, is equal to:

**Choices:**
A. 6 B. 17 C. 9 D. 14

**Correct answer:** D

**Item V147**   $b = \text{-}0.470, a = 351.273, \Delta p = \text{-}1.000$
**Observed correctness:** GPT-5 = 1, GPT-5.2 = 0

**Question:** Let the points $\left(\frac{11}{2}, \alpha\right)$ lie on or inside the triangle with sides $x+y = 11$, $x+2y = 16$, and $2x+3y = 29$. Then the product of the smallest and largest values of $\alpha$ is equal to:

**Choices:**
A. 44 B. C. D. 55

**Correct answer:** C

**Item V323**   $b = \text{-}0.470, a = 351.273, \Delta p = \text{-}1.000$
**Observed correctness:** GPT-5 = 1, GPT-5.2 = 0

**Question:** Let the vectors $a$ and $b$ be of the same magnitude such that $\frac{|a+b|+|a-b|}{|a+b|-|a-b|} = \sqrt{2}+1$. Then $\frac{|a+b|^2}{|a|^2}$ is:

**Choices:**
A. $2 + 4$
sqrt{2} B. $1 +$
sqrt{2} C. $2 +$
sqrt{2} D. $4 + 2$
sqrt{2}

**Correct answer:** C

**Item V327**  $b =$ -0.470,$a = 351.273$,$\Delta p =$ -1.000
**Observed correctness:** GPT-5 = 1, GPT-5.2 = 0

**Question:** Let a random variable X take values 0, 1, 2 and 3 with P(X = 0) = P(X = 1) = p and P(X = 2) = P(X = 3) = $\frac{1-2p}{2}$. If $E(X^2) = 2E(X)$, then the value of $8p - 1$ is:

**Choices:**

A. 0 B. 2 C. 1 D. 3

**Correct answer:** B

**Item V329**  $b =$ -0.470,$a = 351.273$,$\Delta p =$ -1.000
**Observed correctness:** GPT-5 = 1, GPT-5.2 = 0

**Question:** Let $f : \mathbb{R} \to \mathbb{R}$ be a polynomial function of degree four having extreme values at $x = 4$ and $x = 5$. If $\lim\limits_{x \to 0} \dfrac{f(x)}{x^2} = 5$, then $f(2)$ is equal to:

**Choices:**

A. 12 B. 10 C. 8 D. 14

**Correct answer:** B

**Item V333**  $b =$ -0.470,$a = 351.273$,$\Delta p =$ -1.000
**Observed correctness:** GPT-5 = 1, GPT-5.2 = 0

**Question:** Let the length of a latus rectum of an ellipse $\frac{x^2}{a^2} + \frac{y^2}{b^2} = 1$ be 10. If its eccentricity is the minimum value of the function $f(t) = t^2 + t + \frac{11}{12}$, $t \in \mathbb{R}$, then $a^2 + b^2$ is equal to:

**Choices:**

A. 125 B. 126 C. 120 D. 115

**Correct answer:** B

**Item V218**  $b =$ -0.209,$a = 0.000$,$\Delta p =$ -0.000
**Observed correctness:** GPT-5 = 0, GPT-5.2 = 1

**Question:** Let the point $P$ of the focal chord $PQ$ of the parabola $y^2 = 16x$ be $(1, -4)$. If the focus of the parabola divides the chord $PQ$ in the ratio $m : n$, $\gcd(m, n) = 1$, then $m^2 + n^2$ is equal to:

**Choices:**

A. 17 B. 10 C. 37 D. 26

**Correct answer:** A

**Item V243**  $b = 0.689$,$a = 0.000$,$\Delta p =$ -0.000
**Observed correctness:** GPT-5 = 0, GPT-5.2 = 1

**Question:** Each of the angles $\beta$ and $\gamma$ that a given line makes with the positive $y$- and $z$-axes, respectively, is half of the angle that this line makes with the positive $x$-axis. Then the sum of all possible values of the angle $\beta$ is:

**Choices:**

A. $\frac{3\pi}{4}$ B. $\pi$ C. $\frac{\pi}{2}$ D. $\frac{3\pi}{2}$

**Correct answer:** A

**Item V118**  $b = 0.708$,$a = 784.018$,$\Delta p =$ -1.000
**Observed correctness:** GPT-5 = 1, GPT-5.2 = 0

**Question:** Bag 1 contains 4 white balls and 5 black balls, and Bag 2 contains $n$ white balls and 3 black balls. One ball is drawn randomly from Bag 1 and transferred to Bag 2. A ball is then drawn randomly from Bag 2. If the probability that the ball drawn is white is $\frac{29}{45}$, then $n$ is equal to:

**Choices:**
A. 6 B. 3 C. 5 D. 4

**Correct answer:** A

**Item V122**   $b = 0.708, a = 784.018, \Delta p = $ -1.000
**Observed correctness:** GPT-5 = 1, GPT-5.2 = 0

**Question:** Let $O$ be the origin, the point $A$ be $z_1 = \sqrt{3} + 2\sqrt{2}i$, the point $B(z_2)$ be such that $\sqrt{3}|z_2| = |z_1|$ and $\arg(z_2) = \arg(z_1) + \frac{\pi}{2}$. Then

**Choices:**
A. area of triangle $ABO$ is $\frac{11}{\sqrt{3}}$ B. $ABO$ is an obtuse angled isosceles triangle C. area of triangle $ABO$ is $\frac{11}{4}$ D. $ABO$ is a scalene triangle

**Correct answer:** B

**Item V129**   $b = 0.708, a = 784.018, \Delta p = $ -1.000
**Observed correctness:** GPT-5 = 1, GPT-5.2 = 0

**Question:** Two numbers $\mathbf{k}_1$ and $\mathbf{k}_2$ are randomly chosen from the set of natural numbers. Then, the probability that the value of $i^{k_1} + i^{k_2}$, $(i = \sqrt{-1})$ is non-zero, equals

**Choices:**
A. $\frac{1}{2}$ B. $\frac{3}{4}$ C. $\frac{1}{4}$ D. $\frac{2}{3}$

**Correct answer:** B

**Item V379**   $b = 0.708, a = 784.018, \Delta p = $ -1.000
**Observed correctness:** GPT-5 = 1, GPT-5.2 = 0

**Question:** The value of

$$\cot^{-1}\left(\frac{\sqrt{1 + \tan^2(2)} - 1}{\tan(2)}\right) - \cot^{-1}\left(\frac{\sqrt{1 + \tan^2\left(\frac{1}{2}\right)} + 1}{\tan\left(\frac{1}{2}\right)}\right)$$

is equal to:

**Choices:**
A. $\pi - \frac{5}{4}$ B. $\pi - \frac{3}{2}$ C. $\pi + \frac{3}{2}$ D. $\pi + \frac{5}{2}$

**Correct answer:** A

**Item V43**   $b = 0.708, a = 784.018, \Delta p = $ -1.000
**Observed correctness:** GPT-5 = 1, GPT-5.2 = 0

**Question:** Let in a $\triangle ABC$, the length of the side $AC$ be 6, the vertex $B$ be $(1, 2, 3)$ and the vertices $A, C$ lie on the line $\frac{x-6}{3} = \frac{y-7}{2} = \frac{z-7}{2}$. Then the area (in sq. units) of $\triangle ABC$ is:

**Choices:**
A. 17 B. 21 C. 56 D. 42

**Correct answer:** B

**Item V51**   $b = 0.708, a = 784.018, \Delta p = $ -1.000
**Observed correctness:** GPT-5 = 1, GPT-5.2 = 0

**Question:** For a statistical data $x_1, x_2, \ldots, x_{10}$ of 10 values, a student obtained the mean as 5.5 and $\sum_{i=1}^{10} x_i^2 = 371$. He later found that he had noted two values in the data incorrectly as 4 and 5, instead of the correct values 6 and 8, respectively. The variance of the corrected data is

**Choices:**
A. 9 B. 5 C. 7 D. 4

**Correct answer:** C

**Item V160**  $b = 2.440, a = 0.000, \Delta p = $ -0.000
**Observed correctness:** GPT-5 = 0, GPT-5.2 = 1

**Question:** If the equation of the parabola with vertex $\mathbf{V}\left(\frac{3}{2}, 3\right)$ and the directrix $x + 2y = 0$ is $\alpha x^2 + \beta y^2 - \gamma xy - 30x - 60y + 225 = 0$, then $\alpha + \beta + \gamma$ is equal to:

**Choices:**
A. 7 B. 9 C. 8 D. 6

**Correct answer:** B

**Item V194**  $b = 4.702, a = 69.800, \Delta p = $ -0.000
**Observed correctness:** GPT-5 = 0, GPT-5.2 = 0

**Question:** A spherical chocolate ball has a layer of ice-cream of uniform thickness around it. When the thickness of the ice-cream layer is $1\,\mathrm{cm}$, the ice-cream melts at the rate of $81\,\mathrm{cm}^3/\mathrm{min}$ and the thickness of the ice-cream layer decreases at the rate of $\frac{1}{4\pi}\,\mathrm{cm/min}$. The surface area (in $\mathrm{cm}^2$) of the chocolate ball (without the ice-cream layer) is:

**Choices:**
A. $196\pi$ B. $256\pi$ C. $225\pi$ D. $128\pi$

**Correct answer:** B

**Item V343**  $b = 7.214, a = 3.053, \Delta p = $ -0.000
**Observed correctness:** GPT-5 = 0, GPT-5.2 = 0

**Question:** If the shortest distance between the lines $\frac{x-1}{2} = \frac{y-2}{3} = \frac{z-3}{4}$ and $\frac{x}{1} = \frac{y}{\alpha} = \frac{z-5}{1}$ is $\frac{5}{\sqrt{6}}$, then the sum of all possible values of $\alpha$ is:

**Choices:**
A. 3/2 B. -3/2 C. 3 D. -3

**Correct answer:** D

**Item V358**  $b = 7.214, a = 3.053, \Delta p = $ -0.000
**Observed correctness:** GPT-5 = 0, GPT-5.2 = 0

**Question:** Let the system of equations:

$$2x + 3y + 5z = 9,$$
$$7x + 3y - 2z = 8,$$
$$12x + 3y - (4 + \lambda)z = 16 - \mu$$

have infinitely many solutions. Then the radius of the circle centred at $(\lambda, \mu)$ and touching the line $4x = 3y$ is:

**Choices:**
A. $\frac{17}{5}$ B. $\frac{7}{5}$ C. 7 D. $\frac{21}{5}$

**Correct answer:** B

**Item V39**   $b = 7.214, a = 3.053, \Delta p = $ -0.000
**Observed correctness:** GPT-5 = 0, GPT-5.2 = 0

**Question:** The sum of all values of $\theta \in [0, 2\pi]$ satisfying $2\sin^2 \theta = \cos 2\theta$ and $2\cos^2 \theta = 3\sin \theta$ is:

**Choices:**
A. 4
pi B.
frac{5
pi}{6} C.
pi D.
frac{5}{2}

**Correct answer:** C

## C.2   MMLU-Pro (Engineering) selected items

**Item V219**   $b = $ -3.044, $a = 44.856, \Delta p = 0.000$
**Observed correctness:** GPT-5 = 1, GPT-5.2 = 1

**Question:** If a Hexadecimal number needs to convert to binary. For each hexadecimal digit, there will be how many bits

**Choices:**
A. 1 B. 2 C. 4 D. 8

**Correct answer:** C

**Item V231**   $b = $ -3.044, $a = 44.856, \Delta p = 0.000$
**Observed correctness:** GPT-5 = 1, GPT-5.2 = 1

**Question:** A piston and cylinder arrangement has an initial volume of 1.5 ft$\hat{3}$ and contains air at 400psia. The air is then expanded reversibly at a constant temperature of 85°F. What is the work and heat transferred of the process if the final volume of the air inside the cylinder is 4 ft$\hat{3}$. Assume the temperature of the surroundings to be 85°F.

**Choices:**
A. 95 Btu B. 120 Btu C. 108.92 Btu D. 100 Btu

**Correct answer:** C

**Item V255**   $b = $ -3.044, $a = 44.856, \Delta p = 0.000$
**Observed correctness:** GPT-5 = 1, GPT-5.2 = 1

**Question:** A flat piece of plastic, area A = 1 ft$\hat{2}$ and thickness b = 0.252 inch, was tested and found to conduct heat at a rate of 3.0 watts in steady state with temperatures of $T\_0 = 24$°C and $T\_1 = 26$°C on the two main sides of the panel. Find the thermal conductivity of the plastic in cal sec≏1cm≏1 (°K)≏1 at 25°C.

**Choices:**
A. 2.47 × 10≏4 cal sec≏1 cm≏1 (°C)≏1 B. 2.47 × 10≏4 cal sec≏1 cm≏1 (°K)≏1 C. 1.47 × 10≏4 cal sec≏1 cm≏1 (°K)≏1 D. 2.47 × 10≏3 cal sec≏1 cm≏1 (°K)≏1

**Correct answer:** B

**Item V335**   $b = $ -3.044, $a = 44.856, \Delta p = 0.000$
**Observed correctness:** GPT-5 = 1, GPT-5.2 = 1

**Question:** A plate at 60°C, is cooled by forced convection from an airstream. The steady flow velocity of the air stream is observed to be 2m/sec with the temperature and pressure as 27°C and

1 atm., respectively. Find the heat transfer per unit width from the leading edge to a distance of (a) 20 cm, and (b) 40 cm. Also find (c) the drag force over the 40 cm distance.

**Choices:**
A. 75 W, 105 W, 0.004 N B. 81.19 W, 114.8 W, 0.0055 N C. 85 W, 120 W, 0.005 N D. 90 W, 130 W, 0.007 N E. 82 W, 115 W, 0.005 N F. 83 W, 118 W, 0.0065 N G. 86 W, 125 W, 0.0058 N H. 78 W, 112 W, 0.0045 N I. 80 W, 110 W, 0.006 N J. 79.5 W, 108 W, 0.0075 N

**Correct answer:** B

**Item V336** $b = -3.044, a = 44.856, \Delta p = 0.000$
**Observed correctness:** GPT-5 = 1, GPT-5.2 = 1

**Question:** Oil flows through a tube, and heat is transferred to it by forced convection. The tube is 15 ft. long, and its internal diameter measures 0.0303 ft. The flow is measured at 80 lbm/hr, and the wall surface temperature is constant at 350°F. Properties of the oil are: $C\_p = 0.5$ Btu/lbm-°F $k = 0.083$ Btu/hr-ft-°F The viscosity-temperature profile is given by the following data: T(°F) (lbm/ft-hr) 150 15.73 200 12.22 250 9.2 300 6.82 350 4.72 Calculate the heat transfer coefficient and the mean exit temperature of the oil, assuming that is enters the tube at a temperature of 150°F.

**Choices:**
A. Heat transfer coefficient is 21 Btu/hr-ft$\hat{2}$-°F, average outlet temperature is 260°F B. Heat transfer coefficient is 20.05 Btu/hr-ft$\hat{2}$-°F, average outlet temperature is 252.5°F C. Heat transfer coefficient is 18 Btu/hr-ft$\hat{2}$-°F, average outlet temperature is 260°F D. Heat transfer coefficient is 22 Btu/hr-ft$\hat{2}$-°F, average outlet temperature is 245°F

**Correct answer:** B

**Item V229** $b = 0.030, a = 1304.510, \Delta p = 0.000$
**Observed correctness:** GPT-5 = 1, GPT-5.2 = 1

**Question:** Propane is burned in 100% excess air. Combustion is completeand takes place at 100kPa. Determine (a) the dew-pointtemperature of the products of combustion and (b)themoles of H$\_2$O condensed per mole of fuel burned, if theproducts of combustion are cooled to 25°C.

**Choices:**
A. 50°C and 3.0kmolH$\_2$O/kmolC$\_3$H$\_8$ B. 52°C and 2.0 kmol H2O/kmol C3H8 C. 30°C and 3.5 kmol H2O/kmol C3H8 D. 38°C and 2.8 kmol H2O/kmol C3H8 E. 41.53°C and 2.51kmolH$\_2$O/kmolC$\_3$H$\_8$ F. 45°C and 2.3kmolH$\_2$O/kmolC$\_3$H$\_8$ G. 60°C and 1.5 kmol H2O/kmol C3H8 H. 35°C and 2.0kmolH$\_2$O/kmolC$\_3$H$\_8$ I. 48°C and 1.8 kmol H2O/kmol C3H8 J. 55°C and 4.0 kmol H2O/kmol C3H8

**Correct answer:** E

**Item V237** $b = 0.030, a = 1304.510, \Delta p = 0.000$
**Observed correctness:** GPT-5 = 1, GPT-5.2 = 1

**Question:** A 15-hp, 220-volt, 1800-r/min shunt motor has an efficiency of87 per cent at full load. The resistance of the field is 440 ohms.Calculate the (a) Full-load armature current. (b) Torque of the machine.

**Choices:**
A. 54 amps, 38 lb. ft B. 56 amps, 46 lb. ft C. 61 amps, 47 lb. ft D. 58 amps, 44 lb. ft E. 59 amps, 42 lb. ft F. 57.96 amps, 43.77 lb. ft G. 55 amps, 40 lb. ft H. 53.5 amps, 41.5 lb. ft I. 60 amps, 45 lb. ft J. 52 amps, 39 lb. ft

**Correct answer:** F

**Item V251** $b = 0.183, a = 3376.644, \Delta p = 0.000$
**Observed correctness:** GPT-5 = 1, GPT-5.2 = 1

**Question:** Nucleate boiling occurs on a surface at temperature 242°F. If the substance is water, find (a) the maximum heat flux per unit area, and (b) the coefficient of heat transfer.

**Choices:**
A. Maximum heat flux: 250,000 Btu/hr-ft$\hat{2}$, Coefficient of heat transfer: 8,000 Btu/hr-ft$\hat{2}$-°F B. Maximum heat flux: 420,000 Btu/hr-ft$\hat{2}$, Coefficient of heat transfer: 12,500 Btu/hr-ft$\hat{2}$-°F C. Maximum heat flux: 300,000 Btu/hr-ft$\hat{2}$, Coefficient of heat transfer: 10,000 Btu/hr-ft$\hat{2}$-°F D. Maximum heat flux: 350,000 Btu/hr-ft$\hat{2}$, Coefficient of heat transfer: 11,500 Btu/hr-ft$\hat{2}$-°F E. Maximum heat flux: 390,000 Btu/hr-ft$\hat{2}$, Coefficient of heat transfer: 10,800 Btu/hr-ft$\hat{2}$-°F F. Maximum heat flux: 346,729 Btu/hr-ft$\hat{2}$, Coefficient of heat transfer: 11,558 Btu/hr-ft$\hat{2}$-°F G. Maximum heat flux: 410,000 Btu/hr-ft$\hat{2}$, Coefficient of heat transfer: 11,000 Btu/hr-ft$\hat{2}$-°F H. Maximum heat flux: 375,000 Btu/hr-ft$\hat{2}$, Coefficient of heat transfer: 9,500 Btu/hr-ft$\hat{2}$-°F I. Maximum heat flux: 400,000 Btu/hr-ft$\hat{2}$, Coefficient of heat transfer: 12,000 Btu/hr-ft$\hat{2}$-°F J. Maximum heat flux: 265,000 Btu/hr-ft$\hat{2}$, Coefficient of heat transfer: 13,000 Btu/hr-ft$\hat{2}$-°F

**Correct answer:** F

**Item V57**   $b = 0.183, a = 3376.644, \Delta p = 0.000$
**Observed correctness:** GPT-5 = 1, GPT-5.2 = 1

**Question:** A toluene -xylenemixture is fractionated in a bubble-cap column. Approximate the gas-phase mass transfer coefficient. The following pertinent data are given: G_g= vaporflowrate= 23.6kgmole/hr G_l= liquidflowrate= 17.84kgmole/hr M_g = average molecular weight of gas = 93.2 kg/kgmole M_l= average molecular weight of liquid = 93.8 kg/kgmole
rho_l= average density of liquid = 778.4 kg/m$\hat{3}$ t = average column temperature = 113.0°C p = average column pressure = 1 atm.
omega_l0 = liquid velocity at entrance to column = 0.10 m/sec D = inside diameter of column = 680 mm W = weir height above the plate = 65 mm H = weir height above the slots = 30 mm d = outside cap diameter = 85 mm n = number of caps on the column diameter perpendicular to thedirection of liquid flow on the plate = 5 h = liquid head over the weir = 8.1 mm

**Choices:**
A. 4500kgmole/m$\hat{3}$ ullet hr B. 3000kgmole/m$\hat{3}$ ullet hr C. 2500 kgmole/m$\hat{3} \bullet$ hr D. 3552kgmole/m$\hat{3}$ ullet hr E. 2000kgmole/m$\hat{3}$ ullet hr F. 4100 kgmole/m$\hat{3} \bullet$ hr G. 3250 kgmole/m$\hat{3} \bullet$ hr H. 2650 kgmole/m$\hat{3} \bullet$ hr I. 5000 kgmole/m$\hat{3} \bullet$ hr J. 6000 kgmole/m$\hat{3} \bullet$ hr

**Correct answer:** D

**Item V216**   $b = 0.186, a = 3351.787, \Delta p = 0.000$
**Observed correctness:** GPT-5 = 1, GPT-5.2 = 1

**Question:** Use the Laplace transform of f(t) = e$\hat{k}$t,(1) where k is a complex constant of the form k = Rek + i Imk with Rek the real part of k, Imk the imaginary part of k, and i
equiv
surd(- 1), to find the Laplace transforms of f(t) = cosh kt, sinh kt, cos kt, and sin kt.

**Choices:**
A. Lcosh kt = [k / (s$\hat{2}$ - k$\hat{2}$)], Lsinh kt = [s / (s$\hat{2}$ + k$\hat{2}$)], Lcos kt = [k / (s$\hat{2}$ + k$\hat{2}$)], Lsin kt = [s / (s$\hat{2}$ - k$\hat{2}$)] B. Lcosh kt = [2s / (s$\hat{2}$ - k$\hat{2}$)], Lsinh kt = [2k / (s$\hat{2}$ - k$\hat{2}$)], Lcos kt = [2s / (s$\hat{2}$ + k$\hat{2}$)], Lsin kt = [2k / (s$\hat{2}$ + k$\hat{2}$)] C. Lcosh kt = [s / (s$\hat{2}$ + k$\hat{2}$)], Lsinh kt = [k / (s$\hat{2}$ - k$\hat{2}$)], Lcos kt = [s / (s$\hat{2}$ - k$\hat{2}$)], Lsin kt = [k / (s$\hat{2}$ + k$\hat{2}$)]

D. Lcosh kt = [k / (s$\hat{2}$ + k$\hat{2}$)], Lsinh kt = [s / (s$\hat{2}$ + k$\hat{2}$)], Lcos kt = [k / (s$\hat{2}$ - k$\hat{2}$)], Lsin kt = [s / (s$\hat{2}$ + k$\hat{2}$)] E. Lcosh kt = [s / (s$\hat{2}$ - k$\hat{2}$)], Lsinh kt = [2k / (s$\hat{2}$ + k$\hat{2}$)], Lcos kt = [2s / (s$\hat{2}$ - k$\hat{2}$)], Lsin kt = [s / (s$\hat{2}$ + 2k$\hat{2}$)] F. Lcosh kt = [s / (s$\hat{2}$ + k$\hat{2}$)], Lsinh kt = [k / (s $\hat{2}$ + k$\hat{2}$)], Lcos kt = [s / (s$\hat{2}$ - k$\hat{2}$)], Lsin kt = [k / (s$\hat{2}$ - k$\hat{2}$)] G. Lcosh kt = [s / (s$\hat{2}$ + 2k$\hat{2}$)], Lsinh kt = [k / (s$\hat{2}$ + 2k$\hat{2}$)], Lcos kt = [s / (s$\hat{2}$ - 2k$\hat{2}$)], Lsin kt = [k / (s$\hat{2}$ - 2k$\hat{2}$)] H. Lcosh kt = [s / (s$\hat{2}$ - 2k$\hat{2}$)], Lsinh kt = [k / (s$\hat{2}$ + 2k$\hat{2}$)], Lcos kt = [s / (s$\hat{2}$ + 2k$\hat{2}$)], Lsin kt = [k / (s$\hat{2}$ - 2k$\hat{2}$)] I. Lcosh kt = [k / (s$\hat{2}$ + k$\hat{2}$)], Lsinh kt = [s / (s$\hat{2}$ - k$\hat{2}$)], Lcos kt = [s / (s$\hat{2}$ - k$\hat{2}$)], Lsin kt = [k / (s$\hat{2}$ - k$\hat{2}$)] J. Lcosh kt = [s / (s$\hat{2}$ - k$\hat{2}$)], Lsinh kt = [k / (s$\hat{2}$ - k$\hat{2}$)], Lcos kt = [s / (s$\hat{2}$ + k$\hat{2}$)], Lsin kt = [k / (s$\hat{2}$ + k$\hat{2}$)]

**Correct answer:** J

**Item V12**   $b = 0.332, a = 226.860, \Delta p = $ -0.994
**Observed correctness:** GPT-5 = 1, GPT-5.2 = 0

**Question:** What is the pressure inside a boiler which contains liquid water and steam at a temperature of 225°F. Assume that the enthalpy of vaporization of water at 212°F, 1 atm is 970.3 Btu/lbm and the specific volume of the vapor and liquid phases are 26.80 and 0.01672 ft$\hat{3}$/lbm respectively.

**Choices:**
A. 16.25lbf/in$\hat{2}$ B. 17.30lbf/in$\hat{2}$ C. 12.75lbf/in$\hat{2}$ D. 22.50lbf/in$\hat{2}$ E. 15.10lbf/in$\hat{2}$ F. 19.80lbf/in$\hat{2}$ G. 14.50lbf/in$\hat{2}$ H. 18.45lbf/in$\hat{2}$ I. 20.35lbf/in$\hat{2}$ J. 21.65lbf/in$\hat{2}$

**Correct answer:** H

**Item V18**   $b = 0.332, a = 226.860, \Delta p = $ -0.994
**Observed correctness:** GPT-5 = 1, GPT-5.2 = 0

**Question:** A submarine-launched missile, 1 m diameter by 5 m long, is to be studied in a water tunnel to determine the loads acting on it during its underwater launch. The maximum speed during this initial part of the missile's flight is 10 m s≙1. Calculate the mean water tunnel flow velocity if a 1/20 scale model is to be employed and dynamic similarity is to be achieved.

**Choices:**
A. 300 m s≙1 B. 60 m s≙1 C. 500 m s≙1 D. 150 m s≙1 E. 200 m s≙1 F. 100 m s≙1 G. 250 m s≙1 H. 350 m s≙1 I. 50 m s≙1 J. 400 m s≙1

**Correct answer:** E

**Item V25**   $b = 0.332, a = 226.860, \Delta p = $ -0.994
**Observed correctness:** GPT-5 = 1, GPT-5.2 = 0

**Question:** (i)Whatis the circular-milarea of a wire (a) 0.1 in., (b)0.2 in., (c) 0.325 in. in diameter ? (ii) The resistance of a length of copper wire is 3.60 ohms at 20°C.What is its resistance at 80°C ?

**Choices:**
A. (i) (a) 12,500 cir mils, (b) 50,000 cir mils, (c) 126,562 cir mils (ii) 5.60 ohms B. (i) (a) 7,500 cir mils, (b) 30,000 cir mils, (c) 75,625 cir mils (ii) 4.60 ohms C. (i) (a) 15,000 cir mils, (b) 60,000 cir mils, (c) 158,438 cir mils (ii) 6.45 ohms D. (i) (a) 10,000 cir mils, (b) 40,000 cir mils, (c) 105,625 cir mils (ii) 4.45 ohms E. (i) (a) 25,000 cir mils, (b) 100,000 cir mils, (c) 253,125 cir mils (ii) 6.60 ohms F. (i) (a) 8,000 cir mils, (b) 32,000 cir mils, (c) 82,450 cir mils (ii) 7.60 ohms G. (i) (a) 14,000 cir mils, (b) 56,000 cir mils, (c) 140,500 cir mils (ii) 4.75 ohms H. (i) (a) 5,000 cir mils, (b) 20,000 cir mils, (c) 52,812 cir mils (ii) 3.45 ohms I. (i) (a) 18,000 cir mils, (b) 72,000 cir mils, (c) 183,112 cir mils (ii) 5.20 ohms J. (i) (a) 20,000 cir mils, (b) 80,000 cir mils, (c) 210,000 cir mils (ii) 5.45 ohms

**Correct answer:** D

**Item V37**   $b = 0.332, a = 226.860, \Delta p = $ -0.994
**Observed correctness:** GPT-5 = 1, GPT-5.2 = 0

**Question:** What inductance is required to resonate at 10 MHz with a capacitance of 100 pF?

**Choices:**
A. 1.00 $\mu$H B. 1.53 $\mu$H C. 3.14 $\mu$H D. 3.53 $\mu$H E. 0.75 $\mu$H F. 2.00 $\mu$H G. 5.00 $\mu$H H. 4.00 $\mu$H I. 1.77 $\mu$H J. 2.53 $\mu$H

**Correct answer:** J

**Item V43**   $b = 0.332, a = 226.860, \Delta p = $ -0.994
**Observed correctness:** GPT-5 = 1, GPT-5.2 = 0

**Question:** Find the radiation resistance of a single-turn circular loop witha circumference of (1/4) wavelength.

**Choices:**
A. 1.27 ohm B. 1.07 ohm C. 0.107 ohm D. 0.77 ohm E. 0.87 ohm F. 0.57 ohm G. 0.47 ohm H. 0.67 ohm I. 0.97 ohm J. 0.37 ohm

**Correct answer:** D

**Item V45**  $b = 0.332, a = 226.860, \Delta p = $ -0.994
**Observed correctness:** GPT-5 = 1, GPT-5.2 = 0

**Question:** Find the determinate of matrix A A =
vert23
vert
vert12
vert.

**Choices:**
A. 0 B. 2 C. 5 D. -3 E. -1 F. 3 G. 1 H. 4 I. -2 J. 6

**Correct answer:** G

**Item V49**  $b = 0.332, a = 226.860, \Delta p = $ -0.994
**Observed correctness:** GPT-5 = 1, GPT-5.2 = 0

**Question:** A bare steel wire (k = 13 Btu/hr ft°F) is carrying a current of 1000 amp. The diameter of the wire is 0.75 in. The air temperature is 70°F and the surface heat transfer coefficient is 5.0 Btu/hr ft$\hat{2}$ °F. The electrical resistance of the wire per foot length is 0.0001 ohm. Find the centerline temperature of the wire.

**Choices:**
A. 600°F B. 560°F C. 500°F D. 480°F E. 420°F F. 418°F G. 390°F H. 330°F I. 450°F J. 350°F

**Correct answer:** E

**Item V54**  $b = 0.332, a = 226.860, \Delta p = $ -0.994
**Observed correctness:** GPT-5 = 1, GPT-5.2 = 0

**Question:** Consider a screw used in translation along with a 22 mm. insidediameter and 38 mm. outside diameter collar. The screwhas a mean; diameter of 18 mm. and has a single trapezoidal-metric thread cut with a pitch of 4 mm. Determine thetorque required to raise a load of 500 kg, if both the collarand the threads have a coefficient of friction of 0.3.

**Choices:**
A. 45.82 N-m B. 35.82 N-m C. 25.82 N-m D. 55.82 N-m E. 50.82 N-m F. 30.82 N-m G. 42.82 N-m H. 28.82 N-m I. 39.82 N-m J. 60.82 N-m

**Correct answer:** I

**Item V77**  $b = 0.332, a = 226.860, \Delta p = $ -0.994
**Observed correctness:** GPT-5 = 1, GPT-5.2 = 0

**Question:** The solid angle subtended by an area of 2400 cm$\hat{2}$ on the surface of a sphere of diameter 1.2 m is

**Choices:**
A. 3 / 4. B. 2 / 5. C. 1 / 2. D. 1 / 3. E. 1 / 5. F. 1 / 4. G. 2 / 3. H. 4 / 5. I. 3 / 2. J. 5 / 6.

**Correct answer:** G

**Item V81**   $b = 0.332, a = 226.860, \Delta p =$ -0.994
**Observed correctness:** GPT-5 = 1, GPT-5.2 = 0

**Question:** A turbine takes in air at 825K, 690kPaand expands it to 138 kPaand 585K. Assuming air to be an ideal gas, calculate a) the available energy of the air taken in, b) the loss of available energy in the turbine, and c) T_0, the isentropic exit temperature, or, the lowest available temperature.

**Choices:**
A. 290.5 kJ/kg, -75.4 kJ/kg, 515.9K B. 315.5 kJ/kg, -55.4 kJ/kg, 510.9K C. 295.5 kJ/kg, -60.4 kJ/kg, 500.9K D. 280.5 kJ/kg, -65.4 kJ/kg, 540.9K E. 300.5 kJ/kg, -70.4 kJ/kg, 525.9K F. 250.5 kJ/kg, -65.4 kJ/kg, 530.9K G. 305.5 kJ/kg, -60.4 kJ/kg, 520.9K H. 310.5 kJ/kg, -80.4 kJ/kg, 505.9K I. 275.5 kJ/kg, -55.4 kJ/kg, 545.9K J. 320.5 kJ/kg, -50.4 kJ/kg, 535.9K

**Correct answer:** G

**Item V84**   $b = 0.332, a = 226.860, \Delta p =$ -0.994
**Observed correctness:** GPT-5 = 1, GPT-5.2 = 0

**Question:** Express the Boyle temperature of a van der Waals gas in terms of the constants a, b, and R. The final expression for T_B must contain only a, b, and R.

**Choices:**
A. T_B = (a / 2bR) B. T_B = (R / ab) C. T_B = (2a / bR) D. T_B = (aR$\hat{2}$ / b) E. T_B = (bR / a) F. T_B = (b / aR) G. T_B = (a / bR) H. T_B = (a$\hat{2}$ / bR) I. T_B = (a / b)

**Correct answer:** G

**Item V93**   $b = 0.332, a = 226.860, \Delta p =$ -0.994
**Observed correctness:** GPT-5 = 1, GPT-5.2 = 0

**Question:** What is the maximum pressure exerted on an aircraft flying at200 mph at sea level? ( rho = 0.00238 slug/ft$\hat{3}$; p_0 = 14.7 psi.)

**Choices:**
A. 19.5 psia B. 16.42psia C. 14.7psia D. 16.7 psia E. 15.42psia F. 12.7 psia G. 20.1 psia H. 13.89 psia I. 18.34 psia J. 17.5psia

**Correct answer:** E

**Item V455**   $b = 0.361, a = 13654.942, \Delta p =$ -0.000
**Observed correctness:** GPT-5 = 0, GPT-5.2 = 0

**Question:** Four identical alternators each are rated for 20 MVA, 11 KV having a subtransient reactance of 16

**Choices:**
A. 600 MVA B. 800 MVA C. 125 MVA D. 500 MVA E. 700 MVA F. 300 MVA G. 200 MVA H. 400 MVA I. 900 MVA J. 100 MVA

**Correct answer:** D

**Item V566**   $b = 0.361, a = 13654.942, \Delta p =$ -0.000
**Observed correctness:** GPT-5 = 0, GPT-5.2 = 0

**Question:** The surface of a space ship, at temperature 40°F and emissivity0.1, is exposed to space. Consider the space as a perfectblack body at temperature 0°R,and the space ship as agray infinite cylinder. Calculate the net radiant heat loss persquare foot.

**Choices:**
A. 17.2 Btu/hr-ft$\hat{2}$ B. 6.5 Btu/hr-ft$\hat{2}$ C. 14.2 Btu/hr-ft$\hat{2}$ D. 11.5 Btu/hr-ft$\hat{2}$ E. 15.4 Btu/hr-ft$\hat{2}$ F. 9.8 Btu/hr-ft$\hat{2}$ G. 10.71 Btu/hr-ft$\hat{2}$ H. 13.7 Btu/hr-ft$\hat{2}$ I. 12.3 Btu/hr-ft$\hat{2}$ J. 8.5 Btu/hr-ft$\hat{2}$

**Correct answer:** G

## C.3 MMLU-Pro (Law) selected items

**Item V46** $b = -2.436, a = 30.615, \Delta p = 0.000$
**Observed correctness:** GPT-5 = 1, GPT-5.2 = 1

**Question:** A state built a casino and issued bonds to finance its construction. On five occasions, there were episodes of violence in various casinos in the state. The state police attributed the violence to greed and fear at the casinos. To prevent such violence, the state legislature passes a statute prohibiting all gambling at privately owned casinos in the state. Is this law likely to be held constitutional if most casinos in the state were owned by those from out-of-state?

**Choices:**
A. Yes, because the act was expressly authorized by the state legislature. B. Yes, but only if the local interest in safety outweighs the burden of interstate commerce. C. No, because out-of-state casinos are part of interstate commerce. D. No, because the statute violates the due process rights of the owners of the casinos.

**Correct answer:** B

**Item V82** $b = -2.436, a = 30.615, \Delta p = 0.000$
**Observed correctness:** GPT-5 = 1, GPT-5.2 = 1

**Question:** A developer recently acquired a large vacant lot and wished to build a series of 10 tract homes on the lot to offer for sale. On May 1, he telephoned a number of contractors whom he had dealt with in the past and asked them to submit bids to build the homes. One of the contractors in turn sought bids from several subcontractors to determine the overall cost of the job. He notified each subcontractor that he intended to use their bids in submitting a general bid to the developer. A plumber submitted a bid to the contractor in the amount of $50,000, which the contractor used in computing the bid he sent to the developer. On June 1, the contractor submitted a bid to the developer in which he offered to build the 10 tract homes for $500,000. On June 4, the plumber notified the contractor that he had overbooked his crews and could not possibly do the work on the developer's job. On June 5, the developer sent the contractor a signed confirmation letter wherein he stated, ""I hereby accept your offer to build the 10 tract homes for $500,000. ', Thereafter, the contractor hired another plumber to complete the plumbing work at a cost of $60,000. The contractor then explained the situation to the developer and requested that the overall cost of the job be increased to $510,000. The developer refused on the grounds that he had a contract for $500,000 and that this was the contractor's problem. The contractor then built the 10 tract homes, and the developer sent him a check for $500,000. Which of the following best supports a claim for $10,000 by the contractor against the plumber who refused to do the work?

**Choices:**
A. The plumber's bid created an enforceable promissory estoppel situation. B. An option contract was created, because the contractor, with the plumber's knowledge, used the plumber's bid in computing the bid he submitted to the developer. C. The plumber had an implied contract with the contractor, which he violated by refusing to fulfill his bid. D. The plumber had a moral obligation to fulfill his bid as it was already included in the contractor's total cost. E. The contractor relied on the plumber's bid in good faith, creating an enforceable contract. F. The plumber made an offer that the contractor accepted by using the plumber's bid in computing the bid he submitted to the developer. G. The plumber's bid was a binding contract, which the plumber breached by refusing to do the work. H. The contractor made an offer to the plumber that the plumber accepted when he submitted his bid. I. The plumber's bid was a ""firm offer"" that was not revocable, since both the plumber and the contractor were merchants. J. The plumber's bid constituted a unilateral contract which the contractor accepted by submitting his bid to the developer.

**Correct answer:** B

**Item V48**   $b$ = -1.889, $a$ = 21.073, $\Delta p$ = 0.000
**Observed correctness:** GPT-5 = 1, GPT-5.2 = 1

**Question:** Because of a sudden and unanticipated severe shortage of heating fuel, the President has ordered all offices of federal executive agencies to be open only four days per week. The President's order allows an exception to the extent that emergency circumstances require different hours of operation (as in the case of federal hospitals). When Congress enacted the appropriations statute for operating all federal executive agencies, its members assumed that the offices of those agencies would be open five days per week, but Congress did not include such a requirement in its appropriations statute or in any other statute. Is the President's order constitutional?

**Choices:**
A. No, because the heads of the various executive agencies have final responsibility for the operation of those agencies' offices. B. No, because when they passed the statute appropriating monies for the operation of executive agencies, members of Congress assumed that those agencies' offices would be open five days per week. C. Yes, because the Constitution vests the President with plenary authority to direct the administration of all federal agencies in any manner the President deems expedient. D. Yes, because the order relates to the management of the executive branch and is not prohibited by any statute.

**Correct answer:** B

**Item V67**   $b$ = -1.889, $a$ = 21.073, $\Delta p$ = 0.000
**Observed correctness:** GPT-5 = 1, GPT-5.2 = 1

**Question:** A 15-year-old boy was killed during a gang fight. Two days after his funeral, the boy's mother saw a television program about gang violence and was shocked to see video of herself weeping over the boy's body. The video had been shot by the television reporting team while the boy's body was still lying on a public street. The mother suffered severe emotional distress as a result of seeing the video. If the mother sues the television station for invasion of her privacy and that of her son, will the mother be likely to prevail?

**Choices:**
A. No, because the television station has the right to broadcast any footage shot in public. B. No, because the street was open to the public and the subject was newsworthy. C. No, because the broadcast of the video falls under freedom of the press. D. Yes, because the video footage was used without any prior notice to the mother. E. No, because a person has no right to privacy after his or her death. F. Yes, because the television station capitalized on the mother's grief for their program. G. Yes, because the video footage intruded on the mother's private grief. H. Yes, because the mother did not give permission to have the video used in the program. I. Yes, because the mother suffered severe emotional distress as a result of viewing the video.

**Correct answer:** B

**Item V69**   $b$ = -1.889, $a$ = 21.073, $\Delta p$ = 0.000
**Observed correctness:** GPT-5 = 1, GPT-5.2 = 1

**Question:** A defendant decided to rob a bar because he knew that the bar was now utilizing a new security system that was faulty and that nighttime was the best time to break in. Late one night, the defendant broke into the bar and proceeded to load large kegs of beer onto his truck. As he was leaving, the defendant inadvertently threw his cigarette butt into a trash can behind the bar. Soon afterward, the trash ignited into a small fire. Although the defendant had time to douse the fire without any danger to himself, he did not do so. Instead, he climbed into the truck and drove away. Although the fire spread beyond the trashcan, the fire department responded quickly and was able to put it out. The floor behind the bar was charred. The statutes for both burglary and arson in this jurisdiction include residential and business structures. The defendant should be convicted of which of the following crimes?

**Choices:**
A. Robbery and burglary. B. Only Arson. C. Only burglary. D. Arson and larceny. E. Arson,

larceny, and robbery. F. Larceny and burglary. G. Arson, robbery, burglary, and larceny. H. Arson, burglary, and larceny. I. Arson and burglary. J. Arson, robbery, and burglary.

**Correct answer:** H

**Item V76**   $b$ = -1.889,$a$ = 21.073,$\Delta p$ = 0.000
**Observed correctness:** GPT-5 = 1, GPT-5.2 = 1

**Question:** Which of the following will not terminate a contract by operation of law?

**Choices:**
A. Change in law that makes contract impossible B. Bankruptcy of one party C. Expiration of the contract term D. Partial performance E. Supervening illegality F. Death or insanity of a party. G. Fulfillment of contract H. Destruction of the subject matter of contract.

**Correct answer:** D

**Item V92**   $b$ = -0.739,$a$ = 57.469,$\Delta p$ = 0.000
**Observed correctness:** GPT-5 = 1, GPT-5.2 = 1

**Question:** A woman is the fee simple owner of a 1,000-acre tract of land. Half of the tract is swampland totally unfit for cultivation, though it is heavily covered with valuable hardwood timber. The other half is also covered with valuable timber, but it would be land fit for the raising of crops if it were cleared of the timber. The latter section of land is more valuable for cultivation than it is for the growing of timber and has clearings where someone could build a home and live. The woman conveyed the tract to her brother for life. At the time of the conveyance, the swampland had never been used for the production of timber. The brother took possession and cleared 40 acres of the timber on the section that was suitable for cultivation. In addition, the brother cut 60 acres of timber in the swampland, thus becoming the first person to exploit this area. The brother then proceeded to clear some timber on the tract and used it in repairing fences, buildings, equipment, and the like on the property. In an action by the woman to permanently enjoin the brother from cutting any more timber on the swampland section of the tract and to account for profits received in the sale of the timber, the woman will most likely

**Choices:**
A. succeed, because a life tenant may not exploit natural resources where no such prior use had been made. B. not succeed, because a life tenant has a right to make reasonable use of the land. C. succeed, because the life tenant has not obtained explicit permission to cut timber. D. succeed, because a life tenant may not clear timber from swampland. E. not succeed, because the life tenant is allowed to make necessary repairs on the property. F. not succeed, because a life tenant is not liable for permissive waste. G. not succeed, because a life tenant is allowed to exploit natural resources for personal use. H. not succeed, because a life tenant has the right to clear land for cultivation. I. succeed, because a life tenant may not make changes that alter the value of the land. J. succeed, because a life tenant must account for permissive waste to the reversioner or remainderman.

**Correct answer:** A

**Item V23**   $b$ = -0.533,$a$ = 61.504,$\Delta p$ = 0.000
**Observed correctness:** GPT-5 = 1, GPT-5.2 = 1

**Question:** Which statement below best represents Durkheim's view of the function of punishment?

**Choices:**
A. Vengeance. B. Deterrence. C. Restoration. D. Moral education. E. Indemnification. F. Rehabilitation. G. Social control. H. Incapacitation. I. Desert.

**Correct answer:** A

**Item V36**   $b$ = -0.533,$a$ = 61.504,$\Delta p$ = 0.000
**Observed correctness:** GPT-5 = 1, GPT-5.2 = 1

**Question:** Arrests may occasionally be made without a warrant. Which of the following is not one of those circumstances?

**Choices:**
A. A misdemeanor has occurred, and an officer must believe that the person he has arrested committed it. B. An officer must have probable cause that a felony has occurred and that the arrested person committed it. C. An officer may arrest for a felony committed in his presence. D. An officer may arrest for a misdemeanor that is a breach of peace and committed in arresting party's presence.

**Correct answer:** A

**Item V105**   $b = -0.335, a = 2397.262, \Delta p = 0.000$
**Observed correctness:** GPT-5 = 1, GPT-5.2 = 1

**Question:** A homeowner said to a roofer, ""My roof leaks. I think the old tiles are cracked. If you will replace them with all new tiles, I will pay you $5,000. "" The roofer replied, ""Sure, if I can clear my busy schedule. "" The homeowner then remarked, ""That's all right, but let me know soon. "" Three days later, the roofer drove his pickup truck to the homeowner's home and unloaded the materials and equipment needed to perform the roofing job. When the homeowner looked out his window and saw what was transpiring, he immediately ran outside and exclaimed, ""Stop! The deal's off. I decided to repair the roof myself. ""In an action by the roofer against the homeowner for breach of contract, which of the following would provide the roofer with his best theory of recovery?

**Choices:**
A. The homeowner made an offer that proposed a unilateral contract, and the roofer manifested an intent to reject the offer when he began performance by unloading the materials and equipment at the homeowner's house. B. The roofer can recover because the homeowner breached the contract by deciding to repair the roof himself. C. A unilateral contract was formed when the roofer said, ""Sure, if I can clear my busy schedule."" D. A bilateral contract was formed when the roofer said, ""Sure, if I can clear my busy schedule. "" E. A bilateral contract was formed when the homeowner said, ""That's all right, but let me know soon."" F. The homeowner made an offer that proposed a bilateral contract, and the roofer accepted the offer by purchasing the materials and equipment needed for the job. G. The homeowner made an offer that proposed a unilateral contract, and the offer became irrevocable when the roofer purchased the materials and equipment needed for the job. H. A bilateral contract was formed when the roofer purchased the materials and equipment needed to do the job. I. The homeowner made an offer that proposed a unilateral contract, and the roofer manifested an intent to accept the offer when he began performance by unloading the materials and equipment at the homeowner's house. J. The homeowner made an offer that proposed a bilateral contract, and the offer became irrevocable when the roofer unloaded the materials and equipment at the homeowner's house.

**Correct answer:** I

**Item V31**   $b = -0.328, a = 145.265, \Delta p = 0.000$
**Observed correctness:** GPT-5 = 1, GPT-5.2 = 1

**Question:** A man and a woman who were friends decided to buy a car together. The man put $5,000 down and the woman's father co-signed on the loan of $10,000. The title was put in the woman's name and the man made the loan payments. They orally agreed that it was a shared ownership, with equal rights. After several months, their relationship deteriorated. They argued over ownership and use of the car. One day, the man attempted to leave with the car, but the woman climbed in and kept arguing as he was driving away. He then crashed into a pole and destroyed the car just to spite her, saying that if he couldn't have it no one would have it. The authorities arrested the man for auto larceny (stealing a car from the owner) and criminal mischief (damaging property of another or property shared with another). Assuming the accuracy of the foregoing facts beyond a reasonable doubt, which crime or crimes alleged can likely result in a sustainable conviction?

**Choices:**
A. He can be convicted of a different crime not listed. B. He can be convicted of criminal mischief

and a different crime not listed. C. He cannot be convicted of larceny, but he can be convicted of a different crime not listed. D. He cannot be convicted of any crime. E. He can be convicted of both larceny and criminal mischief. F. He can be convicted of larceny but he cannot be convicted of criminal mischief. G. He can be convicted of both larceny and a different crime not listed. H. He can be convicted of criminal mischief only. I. He can be convicted of larceny only. J. He can be convicted of neither larceny nor criminal mischief due to shared ownership.

**Correct answer:** H

**Item V94**   $b$ = -0.305,$a$ = 59.831,$\Delta p$ = 0.000
**Observed correctness:** GPT-5 = 1, GPT-5.2 = 1

**Question:** A defendant was smoking marijuana as he was driving home from work one afternoon. A police officer approached him on the freeway and stopped the car to issue a citation for speeding. As the defendant handed the officer his driver's license, the officer smelled burnt marijuana, saw the joint, and saw an open envelope on the seat next to the defendant containing a substance that looked like marijuana. The officer ordered the defendant out of the car, arrested him for unlawful possession of marijuana. The officer then searched the defendant and found a sealed envelope in the defendant's back pocket. Without asking the defendant's permission, the officer opened the envelope and discovered that it contained cocaine. The officer then unzipped one of the jacket pockets on the jacket the defendant was wearing and found more cocaine and arrested the defendant for possession of cocaine. The defendant was taken to the police station and immediately escorted to an interrogation room. He was given his Miranda warnings, waived his rights, and gave a statement admitting to possession of cocaine and marijuana. Which of the following is a correct statement of the applicable legal principles regarding the search of the defendant?

**Choices:**
A. Any evidence found in the possession of the defendant is admissible, regardless of how it was obtained. B. One who drives an automobile on a public highway does not have a legitimate expectation of privacy. C. If an officer smells marijuana, they are automatically allowed to search the entire vehicle and the person. D. The defendant must give explicit permission for the officer to search their personal belongings, even after an arrest has been made. E. If an individual effectively consents to a search of their person, the evidence gathered during the search is admissible. F. The Fourth Amendment protects individuals from unlawful searches and seizures, making the search and the evidence inadmissible. G. The officer is not permitted to search personal items found on the defendant after the arrest without a warrant. H. The exclusionary rule requires that if an officer conducts an unconstitutional search, the evidence acquired in the course of the officer's subsequent activities is inadmissible. I. The officer is only allowed to search the defendant after they have been formally charged with a crime. J. When a police officer has made a lawful custodial arrest of an individual, he may, as a contemporaneous incident of that arrest, search the person of the individual.

**Correct answer:** J

**Item V355**   $b$ = 0.574,$a$ = 5.217,$\Delta p$ = -0.598
**Observed correctness:** GPT-5 = 1, GPT-5.2 = 0

**Question:** A man outraged by the recent church decision to clear a famous philosopher of charges of heresy, decided to present a lecture, open to the public, disproving the philosopher's theories. A state statute provides that: ""state universities can permit the use of their lecture halls to the public for worthwhile programs of public benefit, upon approval of the school board. ""The appropriate school board refused to make a university lecture hall available to the man on the grounds that the proposed lecture was not of worthwhile benefit to the public. As a result, the man brought suit in a state court against the school board and requested injunctive relief requiring the board to allow him the use of the lecture hall. The trial court denied relief and dismissed the suit. The judgment was affirmed by the state appellate court, and is now before the U. S. Supreme Court. In analyzing the state statute, which of the following statements is least accurate?

**Choices:**
A. The statute is unconstitutionally overbroad, because it may result in the exclusion of protected

speech as well as unprotected speech. B. The statute, as applied to the man, does not violate his First Amendment rights because his proposed speech is not political and, therefore, not among the classes of speech that are protected. C. Indirect speech, regulations are only permissible if necessary to serve compelling state interests. D. The statute is a prior restraint on speech, which unconstitutionally vests unfettered discretion in the school board to decide who may use university lecture halls.

**Correct answer:** B

**Item V296**   $b = 0.994, a = 17.720, \Delta p = $ -0.996
**Observed correctness:** GPT-5 = 1, GPT-5.2 = 0

**Question:** Billy was charged with grand theft. The trial began on a Thursday afternoon. The jury was empaneled, sworn and released for the day. Since Friday was the Fourth of July, the judge asked the jurors to return on Monday. The trial began again on Monday morning at 8:30. By late evening the judge had instructed the jury. Due to the lateness of the hour, the jurors were sequestered for the evening to allow them to get an early start the next morning. The jurors returned Tuesday morning and were unable to reach a verdict. Unable to reach a verdict, the trial judge allowed the jurors to go home that evening. On Wednesday morning, the jury assembled and returned a verdict of guilty. On appeal, which of the following is Billy's strongest issue for seeking a reversal?

**Choices:**
A. The fact that the jury was allowed to go home after being sworn. B. The fact that the jurors did not begin to consider evidence until several days after they were empaneled. C. The fact that the jury was allowed to go home after they began deliberations. D. The fact that the trial began on a holiday weekend. E. The fact that the jury was not given enough time to deliberate. F. The fact that the jury was sequestered for only one night. G. The fact that the jury took several days to return a verdict. H. The fact that the jury returned a verdict of guilty. I. The fact that the jury did not start deliberating until the following week. J. The fact that the trial began late in the afternoon.

**Correct answer:** C

**Item V370**   $b = 0.994, a = 17.720, \Delta p = $ -0.996
**Observed correctness:** GPT-5 = 1, GPT-5.2 = 0

**Question:** A man owned a piece of land with a run-down log cabin on it. He lost interest in it, letting tax bills pile up and failing to maintain it. In 1990, a homeless couple occupied the crumbling cabin. They restored it, fenced the perimeter, and maintained the grounds. They started paying the taxes, despite knowing there was an absentee owner. In 2000, the owner appeared and demanded that they pay rent. They agreed but stopped the payments two years later after consulting an attorney. The required statutory period in the state is 20 years. In 2011 the owner filed an ejectment action, asserting his ownership rights. Leaving aside any other theories or equities, can the couple prevail on a theory of adverse possession?

**Choices:**
A. No, because the statutory period had not been met when the owner filed for ejectment. B. Yes, because they maintained the grounds of the property. C. Yes, because they had exclusive possession of the property. D. Yes, because they were on the land for over 20 years. E. Yes, because they owned the property due to paying the taxes on it. F. No, because they missed some of the tax payments during that 20-year period. G. No, because they knew there was an absentee owner. H. Yes, because they made improvements to the property. I. No, because the hostile possession was interrupted for two years when they paid rent. J. No, because they did not have a legal claim to the property.

**Correct answer:** I

**Item V376**   $b = 0.994, a = 17.720, \Delta p = $ -0.996
**Observed correctness:** GPT-5 = 1, GPT-5.2 = 0

**Question:** Two cars were driving within the legal speed limit as they approached an intersection. There are no traffic lights, stop signs, or warnings posted at this intersection. Both vehicles entered the intersection simultaneously and collided. In the accident, one of the drivers suffered a broken pelvis and internal injuries. Although the other driver's truck was slightly damaged, he did not suffer any physical injuries. Thereafter, the injured driver sued the city for negligence, claiming that the city failed to provide adequate warning at the intersection, which she contended was particularly dangerous and heavily traveled. At trial, the city attorney offers evidence that the intersection has been the same for 14 years, and no other accidents have occurred during that time. Upon objection by the injured driver's attorney, this evidence should be

**Choices:**
A. excluded, because it is improper negative evidence. B. admitted as relevant evidence of a fact of consequence to the action. C. admitted, as it is critical for the jury to understand the history of the intersection. D. excluded, because the dangers of unfair prejudice and confusion of the issues may tend to mislead the jury. E. admitted, but only if the city can prove that there have been no significant changes in traffic patterns in the last 14 years. F. excluded, because it can potentially bias the jury against the injured driver. G. excluded, because it is irrelevant to the current accident. H. admitted, as it shows the city's negligence in maintaining the intersection. I. excluded, as it is not directly related to the case at hand. J. admitted, provided that the jury is instructed about the danger of inferential negative evidence.

**Correct answer:** B

**Item V39**  $b = 0.994, a = 17.720, \Delta p = -0.996$
**Observed correctness:** GPT-5 = 1, GPT-5.2 = 0

**Question:** A landlord is the owner in fee simple of a tract of land on which is situated a large office building. The landlord leases the land and building thereon to a tenant for a term of seven years, commencing on August 1, 2001, and terminating at midnight on July 31, 2008. The lease contains the following provisions:""(1) The tenant covenants to pay the rent of $750 per month on the first day of each month. ""After three years of the lease had expired, the tenant assigned the entire balance of the lease period to an attorney, who took immediate possession of the leased property. Then in 2005, the attorney assigned his leasehold interest to a doctor. The doctor went into possession, but failed to pay any rent for two months. After the doctor failed to make his rental payments for the first two months of his lease, the landlord brought suit against the tenant to recover for the unpaid rent. Judgment should be for

**Choices:**
A. the tenant, because the landlord did not formally approve the assignments. B. the landlord, because the tenant's contractual obligation under the lease survived the assignments. C. the landlord, because the tenant and the assignees breached the lease agreement. D. the landlord, because the doctor, as the final assignee, failed to fulfill his obligation to pay rent. E. the tenant, because he was not directly responsible for the doctor's failure to pay rent. F. the landlord, because he did not object to the assignments. G. the landlord, because the attorney failed to pay rent for two months. H. the tenant, because his assignment to the attorney constituted a novation, thereby extinguishing his obligation to pay rent. I. the tenant, because the landlord did not take action when the doctor failed to pay rent. J. the tenant, because the doctor, as assignee, would only be held liable.

**Correct answer:** B

**Item V438**  $b = 0.994, a = 17.720, \Delta p = -0.996$
**Observed correctness:** GPT-5 = 1, GPT-5.2 = 0

**Question:** A customer at a fish market was leaving the store after purchasing an assortment of shrimp, oysters, and scallops. He was walking along the sidewalk in front of the store when he slipped on a piece of eel. He brought suit against the owner of the market claiming that he suffered leg and back injuries. The owner, although admitting that the customer was injured by slipping on the eel, denied negligence and claimed that the customer was contributorily negligent. At trial, the owner calls a witness to testify that before the fall he heard someone call out to the customer, ""Watch it, buddy, you're going to step on that piece of fish. ""The witness's testimony is

**Choices:**
A. admissible, as it is a statement that was made in the heat of the moment. B. inadmissible, because the customer did not acknowledge hearing the warning. C. inadmissible, because it was not a spontaneous utterance. D. inadmissible, as hearsay not within any exception. E. admissible, because it is relevant as to the customer's contributory negligence. F. admissible, because it provides context to the event. G. inadmissible, because the witness was not directly involved in the incident. H. admissible, because it indicates that the owner had no control over the situation. I. admissible, as a statement of the declarant's present state of mind. J. inadmissible, as it is considered second-hand information.

**Correct answer:** E

**Item V582** $b = 0.994, a = 17.720, \Delta p = $ -0.996
**Observed correctness:** GPT-5 = 1, GPT-5.2 = 0

**Question:** A town had been experiencing a rash of bank robberies. All the witnesses at each bank had described a man with a gun who said, ""Get down on the floor and no one has to die today!"" Following a series of leads, the detectives in the case finally made an arrest. At the police station, the suspect was placed in a lineup. One of the victims observed the lineup, and after each of the six participants stated, ""Get down on the floor and no one has to die today,"" the victim identified the suspect as the perpetrator. The defendant's attorney subsequently moved to suppress the out-of-court identification, arguing that because the defendant was forced to speak, this violated his Fifth Amendment right to remain silent, which had been invoked previously by the defendant. The police requirement that the defendant state at the lineup, ""Get down on the floor and no one has to die today""

**Choices:**
A. was a permissible police procedure as the statement was not incriminating. B. violated the defendant's right to a fair trial. C. was an infringement on the defendant's right to counsel. D. was not a violation because the defendant had already invoked his right to remain silent. E. would not prohibit the defendant's lineup identification from being introduced at trial. F. would make the defendant's lineup identification admissible only if corroborated by other evidence. G. invalidated the lineup. H. tainted the victim's identification of the defendant. I. was an acceptable method of identification because the defendant was not under oath.

**Correct answer:** E

**Item V85** $b = 0.994, a = 17.720, \Delta p = $ -0.996
**Observed correctness:** GPT-5 = 1, GPT-5.2 = 0

**Question:** A state has a hit-and-run statute in effect that makes it a crime to leave the scene of an accident. One afternoon, a 9-year-old victim was riding her bicycle along the street. As the victim swerved into the southbound lane, her bicycle was struck by a car driven by the defendant. The victim was knocked off her bike and thrown onto the sidewalk adjacent to the street. Although the victim received some minor scrapes and bruises, she was not seriously injured. Following the accident, the car driven by the defendant sped away. Moments later, however, a tractor-trailer crashed into the rear of a car about 30 feet from where the victim was lying. The car almost instantly caught fire as its gas tank exploded. The victim, who was engulfed in the flaming wreckage, was killed. If the defendant is charged with involuntary manslaughter for the death of the victim, the defendant should be found

**Choices:**
A. guilty, because she failed to report the accident to the authorities. B. guilty, because she unlawfully fled the scene of an accident in violation of her statutory duty. C. guilty, because she should have anticipated the potential for further accidents. D. guilty, because her actions directly resulted in the victim's death. E. not guilty, because there was not a sufficient causal connection between her actions and the victim's death to impose criminal responsibility. F. guilty, because her failure to render aid to the victim would make the defendant criminally responsible for the victim's death.

**Correct answer:** E

**Item V878**    $b = 0.994, a = 17.720, \Delta p = $ -0.996
**Observed correctness:** GPT-5 = 1, GPT-5.2 = 0

**Question:** In what way is Responsibility to Protect (R2P) different from humanitarian intervention?

**Choices:**
A. R2P is less arbitrary because it requires some UNSC input and its primary objective is to avert a humanitarian crisis B. R2P always involves economic sanctions, whereas humanitarian intervention does not C. R2P is essentially the same as humanitarian intervention D. R2P requires a call for assistance by the State in distress E. R2P always involves armed force, whereas humanitarian intervention does not F. Humanitarian intervention is essentially the same as R2P G. Humanitarian intervention always requires UNSC approval while R2P does not H. Humanitarian intervention requires a call for assistance by the State in distress while R2P does not I. R2P is more arbitrary because it does not require UNSC input J. R2P is a broader concept that includes economic sanctions, whereas humanitarian intervention is strictly military

**Correct answer:** A

**Item V933**    $b = 0.994, a = 17.720, \Delta p = $ -0.996
**Observed correctness:** GPT-5 = 1, GPT-5.2 = 0

**Question:** A state enacts a statute that prohibits ""anyone over60 years of age to run for public office. "" A state senator has been in office for three terms and wishes to seek re-election. The senator, who is 61, brings suit challenging the constitutionality of the state statute. Which of the following best states the burden of persuasion?

**Choices:**
A. The petitioner must show the statute violates due process by depriving her of the right to be a candidate. B. The state needs to show that the statute is rationally related to a compelling government interest. C. Since a fundamental right is involved, the petitioner must show the regulation is necessary to vindicate a compelling government interest. D. Since no fundamental right is involved, the state must show the age restriction is essential to maintain a legitimate government interest. E. The state must prove that the age restriction is essential to maintain the efficiency of the government. F. The petitioner must show the statute violates the fundamental right of freedom of expression. G. Since no fundamental right is involved, the petitioner must show the age restriction is not rationally related to a legitimate government interest.

**Correct answer:** G

**Item V947**    $b = 0.994, a = 17.720, \Delta p = $ -0.996
**Observed correctness:** GPT-5 = 1, GPT-5.2 = 0

**Question:** A farmer has a large field where he grows corn. He hires a crop-duster to spray the field, but the crop- duster mistakenly sprays a strong herbicide, rather than a pesticide, and kills all the corn plants in the field. Several days later, a construction company building a house near the farmer's field negligently hits a gas line and starts a fire that burns everything in the area, including the farmer's field. If the farmer brings negligence claims against the crop-duster and the construction company in order to recover damages for the destruction of his corn crop, the most likely result is

**Choices:**
A. neither the crop-duster nor the construction company is liable, because neither is a ""but for"" cause of the harm. B. both the crop-duster and the construction company are liable, because each engaged in a negligent action that was sufficient to destroy the farmer's corn crop. C. only the crop-duster is liable, because its negligence occurred before the construction company's negligence. D. both the crop-duster and the construction company are liable, because they both contributed to the loss of the farmer's corn crop. E. neither the crop-duster nor the construction company is liable, because the farmer did not have insurance for his corn crop. F. neither the crop-duster nor the construction company is liable, because the farmer failed to protect his field. G. only the

construction company is liable, because the fire caused more damage than the herbicide. H. only the crop-duster is liable, because the construction company's fire was an unforeseeable event. I. only the construction company is liable, because its negligence would have destroyed the farmer's corn crop no matter what the crop- duster did. J. the crop-duster is liable for the value of the corn crop and the construction company is liable for the damage to the field.

**Correct answer:** C

**Item V968**   $b = 0.994, a = 17.720, \Delta p = -0.996$
**Observed correctness:** GPT-5 = 1, GPT-5.2 = 0

**Question:** What is the protective principle of jurisdiction?

**Choices:**
A. It is jurisdiction based on the nationality of the victim B. It is jurisdiction in order to protect state sovereignty C. It is jurisdiction based on the severity of the crime committed D. It is jurisdiction in order to uphold international law E. It is jurisdiction based on the harm to national interests by conduct committed abroad F. It is jurisdiction based on the location of the crime committed G. It is jurisdiction based on the nationality of the offender H. It is jurisdiction in order to protect international human rights I. It is jurisdiction to protect the interests of foreign nationals within the country J. It is jurisdiction in order to protect one's nationals abroad

**Correct answer:** E

## C.4   MMLU-Pro (Math) selected items

**Item V27**   $b = -2.732, a = 85.589, \Delta p = 0.000$
**Observed correctness:** GPT-5 = 1, GPT-5.2 = 1

**Question:** All of Bella's photos are in 8 photo books. Each photo book has 25 photos. Which expression shows how to find the total number of photos Bella has?

**Choices:**
A. $25 \times 8$ B. $8 + 25$ C. $25 + 25$ D. $25 \times 25$ E. $8 - 25$ F. $25 - 8$ G. $25 + 8$ H. $8 \div 25$ I. $25 \div 8$ J. $8 \times 8$

**Correct answer:** A

**Item V101**   $b = -2.516, a = 40.915, \Delta p = 0.000$
**Observed correctness:** GPT-5 = 1, GPT-5.2 = 1

**Question:** The set of all nth roots of unity under multiplication of complex numbers form a/an

**Choices:**
A. non-abelian group B. infinite group C. commutative semigroups with identity D. non-commutative group E. abelian group F. non-commutative semi group with identity G. semi group with identity

**Correct answer:** E

**Item V116**   $b = -2.516, a = 40.915, \Delta p = 0.000$
**Observed correctness:** GPT-5 = 1, GPT-5.2 = 1

**Question:** A dentist has noticed that about two children in every seven whom he sees professionally develop cavities before they turn 10 years old. Last week he examined the teeth of five unrelated children younger than 10. Let X be the number of children who develop cavities before turning 10. Which of the following gives the probability that at least one will develop a cavity before turning 10?

**Choices:**
A. P(X = 3 out of 5) B. P(X = 0) C. P(X = 7 out of 10) D. P(X = 1) E. 1 − P(X = 1) F. P(X =

1, 2, 3, 4, 5) G. P(X = 2 out of 7) H. P(X = 2, 3, 4, 5, 6, 7) I. 1 − P(X = 0) J. P(X = 2 out of 5)

**Correct answer:** I

**Item V125** $b$ = -2.516,$a$ = 40.915,$\Delta p$ = 0.000
**Observed correctness:** GPT-5 = 1, GPT-5.2 = 1

**Question:** What is the least common multiple of 4 and 10?

**Choices:**
A. 100 B. 20 C. 14 D. 40 E. 80 F. 60 G. 70 H. 50 I. 200 J. 30

**Correct answer:** B

**Item V137** $b$ = -2.516,$a$ = 40.915,$\Delta p$ = 0.000
**Observed correctness:** GPT-5 = 1, GPT-5.2 = 1

**Question:** Order the numbers 6, -1, 8, -3, -4 from least to greatest.

**Choices:**
A. -4, 8, -1, -3, 6 B. -4, -3, -1, 6, 8 C. 6, -1, 8, -3, -4 D. -4, -1, -3, 6, 8 E. -1, 6, -3, -4, 8 F. -3, -4, -1, 8, 6 G. -1, -3, -4, 6, 8 H. -3, -4, 6, -1, 8 I. 6, 8, -1, -3, -4 J. 8, 6, -1, -3, -4

**Correct answer:** B

**Item V28** $b$ = -2.516,$a$ = 40.915,$\Delta p$ = 0.000
**Observed correctness:** GPT-5 = 1, GPT-5.2 = 1

**Question:** Which number written in exponential notation is equivalent to 343?

**Choices:**
A. $7\hat{4}9$ B. $7\hat{3}$ C. $49\hat{3}$ D. $2944\hat{9}$ E. $49\hat{7}$ F. $3\hat{7}$ G. $1\hat{3}43$ H. $7\hat{7}$ I. $3\hat{4}9$

**Correct answer:** B

**Item V33** $b$ = -2.516,$a$ = 40.915,$\Delta p$ = 0.000
**Observed correctness:** GPT-5 = 1, GPT-5.2 = 1

**Question:** A tournament has 393 players with 3 players on each team. Which expression shows how to find the number of teams in the tournament?

**Choices:**
A. 3 + 393 B. 3 × 393 C. (393 - 3) × 3 D. 393 ÷ 3 E. (393 + 3) ÷ 3 F. 393 × 3 G. 393 + 3 H. 393 − 3 I. 3 - 393 J. 3 ÷ 393

**Correct answer:** D

**Item V4** $b$ = -2.516,$a$ = 40.915,$\Delta p$ = 0.000
**Observed correctness:** GPT-5 = 1, GPT-5.2 = 1

**Question:** Statement 1 | Any set of two vectors in R$\hat{2}$ is linearly independent. Statement 2 | If V = span(v1, ... , vk) and v1, ... , vk are linearly independent, then dim(V) = k.

**Choices:**
A. False, False B. True, False C. False, True D. Cannot determine, False E. True, True F. Cannot determine, Cannot determine G. False, Cannot determine H. True, Cannot determine I. Cannot determine, True J. False, False, but there are exceptions

**Correct answer:** C

**Item V9** $b$ = -2.516,$a$ = 40.915,$\Delta p$ = 0.000
**Observed correctness:** GPT-5 = 1, GPT-5.2 = 1

**Question:** Statement 1 | There exists a free abelian group of every positive integer rank. Statement 2 | A finitely generated abelian group is free abelian if its Betti number equals the number of elements in some generating set.

**Choices:**
A. False, False B. True, Cannot be determined C. False, True D. Cannot be determined, True E. False, Cannot be determined F. Statement 1 is false for certain positive integers, Statement 2 is false G. True, False H. Statement 1 is true only for certain positive integers, Statement 2 is true I. Cannot be determined, False J. True, True

**Correct answer:** J

**Item V105**   $b$ = -0.457, $a$ = 7226.276, $\Delta p$ = 0.000
**Observed correctness:** GPT-5 = 1, GPT-5.2 = 1

**Question:** Statement 1 | A homomorphism may have an empty kernel. Statement 2 | It is not possible to have a nontrivial homomorphism of some finite group into some infinite group.

**Choices:**
A. Statement 1 is true, Statement 2 is not given B. Both statements are true, but they contradict each other C. Both statements are not given D. False, True E. Statement 1 is not given, Statement 2 is true F. Statement 1 is false, Statement 2 is not given G. False, False H. True, True I. True, False J. Statement 1 is not given, Statement 2 is false

**Correct answer:** G

**Item V109**   $b$ = -0.457, $a$ = 7226.276, $\Delta p$ = 0.000
**Observed correctness:** GPT-5 = 1, GPT-5.2 = 1

**Question:** For T: Z x Z -> Z where T(1, 0) = 3 and T(0, 1) = -5, find T(-3,2).

**Choices:**
A. -20 B. 10 C. -15 D. 5 E. -5 F. 15 G. 19 H. -10 I. -19 J. 20

**Correct answer:** I

**Item V36**   $b$ = -0.457, $a$ = 7226.276, $\Delta p$ = 0.000
**Observed correctness:** GPT-5 = 1, GPT-5.2 = 1

**Question:** A tank is in the shape of a triangular prism. If the triangular base has an area of 116 square feet, and the tank is 30 feet tall, how much water would the tank contain when it is full?

**Choices:**
A. 2320 ft$\hat{3}$ B. 8700 ft$\hat{3}$ C. 2900 ft$\hat{3}$ D. 1725 ft$\hat{3}$ E. 5000 ft$\hat{3}$ F. 3480 ft$\hat{3}$ G. 1740 ft$\hat{3}$ H. 6960 ft$\hat{3}$ I. 4640 ft$\hat{3}$ J. 4000 ft$\hat{3}$

**Correct answer:** F

**Item V1017**   $b$ = 3.828, $a$ = 15.157, $\Delta p$ = -1.000
**Observed correctness:** GPT-5 = 1, GPT-5.2 = 0

**Question:** The region bounded by the curves y = x and y = x$\hat{2}$ in the first quadrant of the xy-plane is rotated about the y-axis. The volume of the resulting solid of revolution is

**Choices:**
A. $2\pi$ B. $3\pi/2$ C. $2\pi/3$ D. $3\pi/4$ E. $\pi$ F. $\pi/12$ G. $\pi/2$ H. $\pi/3$ I. $\pi/4$ J. $\pi/6$

**Correct answer:** J

**Item V1025**   $b$ = 3.828, $a$ = 15.157, $\Delta p$ = -1.000
**Observed correctness:** GPT-5 = 1, GPT-5.2 = 0

**Question:** Find the remainder when $25\hat{1}059$ is divided by 23.

**Choices:**
A. 22 B. 6 C. 11 D. 5 E. 13 F. 3 G. 12 H. 19 I. 8 J. 16

**Correct answer:** I

**Item V1032** $b = 3.828, a = 15.157, \Delta p = $ -1.000
**Observed correctness:** GPT-5 = 1, GPT-5.2 = 0

**Question:** For the complex number z = a + bi, it is known that a and b are positive integers, and that $z\hat{3} + z\hat{2} + z$ is a real number. Find the smallest possible value for b.

**Choices:**
A. 6 B. 9 C. 1 D. 8 E. 4 F. 2 G. 11 H. 10 I. 7 J. 5

**Correct answer:** G

**Item V936** $b = 3.828, a = 15.157, \Delta p = $ -1.000
**Observed correctness:** GPT-5 = 1, GPT-5.2 = 0

**Question:** Let {X_n: n
geq 1} be independent, identically distributed random variables taking integer values 1,-1. Let S_0=0, S_n=
sum_i=1ñ X_i. Let P(X_i=1)=0.8 and P(X_i=-1)=0.2. The range R_n of S_0,S_1,...,S_n is the number of distinct values taken by the sequence. Then what is the limit of n⁻1E[R_n] as n goes to infinity? Here E[R_n] is the expectation over the random variable R_n.

**Choices:**
A. 1.0 B. 0.5 C. 1.2 D. 1.6 E. 0.8 F. 0.9 G. 1.4 H. 0.2 I. 0.6 J. 0.4

**Correct answer:** I

**Item V972** $b = 3.828, a = 15.157, \Delta p = $ -1.000
**Observed correctness:** GPT-5 = 1, GPT-5.2 = 0

**Question:** Which of the following statements is true?

**Choices:**
A. Empty relation _ is not reflexive. B. Number of relations from A = x, y, z to B= (1, 2, 3), is 64. C. Properties of a relation being symmetric and being un-symmetric are positive of each other. D. Empty relation _ is reflexive E. Properties of a relation being symmetric and being un-symmetric are negative of each other. F. Number of relations form A = x, y, z to B= (1, 2), is 64. G. Every equivalence relation is a partial-ordering relation. H. Every partial-ordering relation is an equivalence relation. I. Number of relations from A = x, y, z to B= (1, 2), is 32. J. Properties of a relation being symmetric and being anti-symmetric are negative of each other.

**Correct answer:** F

**Item V976** $b = 3.828, a = 15.157, \Delta p = $ -1.000
**Observed correctness:** GPT-5 = 1, GPT-5.2 = 0

**Question:** A one-sided test of a hypothesis about a population mean, based on a sample of size 14, yields a P-value of 0.075. Which of the following best describes the range of tvalues that would have given this P-value?

**Choices:**
A. $1.000 < $ t$< 1.500$ B. $1.500 < $ t$< 2.000$ C. $1.771 < $ t$< 2.160$ D. $1.771 < $ t$< 2.500$ E. $1.782 < $ t$< 2.150$ F. $1.350 < $ t$< 1.771$ G. $1.345 < $ t$< 1.761$ H. $1.761 < $ t$< 2.160$ I. $1.356 < $ t$< 1.782$ J. $1.200 < $ t$< 1.400$

**Correct answer:** F

**Item V980**  $b = 3.828, a = 15.157, \Delta p = $ -1.000
**Observed correctness:** GPT-5 = 1, GPT-5.2 = 0

**Question:** During the years 1886 through 2000 there were an average of 8.7 tropical cyclones per year, of which an average of 5.1 became hurricanes. Assuming that the probability of any cyclone becoming a hurricane is independent of what happens to any other cyclone, if there are five cyclones in one year, what is the probability that at least three become hurricanes?

**Choices:**
A. 0.586 B. 0.345 C. 0.621 D. 0.313 E. 0.528 F. 0.412 G. 0.701 H. 0.274 I. 0.658 J. 0.789

**Correct answer:** I

**Item V981**  $b = 3.828, a = 15.157, \Delta p = $ -1.000
**Observed correctness:** GPT-5 = 1, GPT-5.2 = 0

**Question:** Taxicabs in a metropolitan area are driven an average of 75,000 miles per year with a standard deviation of 12,000 miles. What is the probability that a randomly selected cab has been driven less than 100,000 miles if it is known that it has been driven over 80,000 miles? Assume a normal distribution of miles per year among cabs.

**Choices:**
A. 0.72 B. 0.45 C. 0.66 D. 0.80 E. 0.94 F. 0.06 G. 0.20 H. 0.34 I. 0.50 J. 0.86

**Correct answer:** E

**Item V985**  $b = 3.828, a = 15.157, \Delta p = $ -1.000
**Observed correctness:** GPT-5 = 1, GPT-5.2 = 0

**Question:** Assume that the masses of chicken eggs are normally distributed with a mean of 45 g and a standard deviation of 4 g. What mass of egg would be the 25th percentile of the masses of all the eggs?

**Choices:**
A. 42.8 g B. 42.7 g C. 42.5 g D. 42.0 g E. 42.3 g F. 42.6 g G. 42.2 g H. 42.1 g I. 42.4 g J. 41.9 g

**Correct answer:** E

**Item V996**  $b = 3.828, a = 15.157, \Delta p = $ -1.000
**Observed correctness:** GPT-5 = 1, GPT-5.2 = 0

**Question:** There are 2,817 homes in the town of West Valley. Each home uses an average of 380 gallons of water each day. Use the expression below to find the total number of gallons of water the homes in West Valley use on average each day. 2,817 × 380 What is the total number of gallons of water the homes in West Valley use on average each day?

**Choices:**
A. 860,460 gallons B. 1,060,460 gallons C. 1,150,460 gallons D. 1,070,460 gallons E. 870,460 gallons F. 900,460 gallons G. 980,460 gallons H. 1,080,460 gallons I. 1,090,460 gallons J. 1,100,460 gallons

**Correct answer:** D

**Item V997**  $b = 3.828, a = 15.157, \Delta p = $ -1.000
**Observed correctness:** GPT-5 = 1, GPT-5.2 = 0

**Question:** The state of Georgia has 159 counties and covers 58056 square miles. What is the average number of square miles per county? Round your answer to the nearest whole number.

**Choices:**
A. 318 square miles per county B. 150 square miles per county C. 250 square miles per county D. 350 square miles per county E. 500 square miles per county F. 274 square miles per county G. 200

square miles per county H. 159 square miles per county I. 365 square miles per county J. 400 square miles per county

**Correct answer:** I

**Item V66**   $b = 6.619, a = 36.876, \Delta p$ = -1.000
**Observed correctness:** GPT-5 = 1, GPT-5.2 = 0

**Question:** Suppose $f(x)$ is an invertible function, and suppose that $f(2) = f^{-1}(2) = 4$. What is the value of $f(f(2))$?

**Choices:**
A. $f(4)$ B. $f(f^{-1}(2))$ C. 1 D. $f^{-1}(4)$ E. $f^{-1}(2)$ F. 8 G. 16 H. 0 I. 2 J. 4

**Correct answer:** I

