# OpenReview forum: "Longitudinal Evaluation of Large Language Models"
_TMLR — Under review for TMLR_

### Review · Reviewer_u2qW · 2026-06-09

**Summary Of Contributions:**

The paper proposes an IRT-based, 2PL model for LLM evaluation and item functioning analysis. A novelty is that instead of estimating a fix ability parameter for each model separately, the authors propose to authoregressively estimate the ability of each model generation depending on the previous model generation ability. They apply their framework to different model families, most significantly on a timeline of 18 GPT models.

**Audience:**

Yes

**Audience Explanation:**

I do think that "IRT-like" models could help us to learn more about benchmarks and changes between model generations. I think IRT is very popular, especially for efficient benchmarking and this paper could be interesting to this community.

**Broader Impact Concerns:**

I don't think that there are any ethical concerns with this work that would require a Broader Impact Statement.

**Claims And Evidence:**

No

**Claims Explanation:**

The paper proposes a new estimation method without any quality analysis, fit metrics or item residuals. There is also no comparison to alternative methods or a comparison to purely accuracy-based methods. Furthermore, there was no discussion of whether the chosen datasets actually measure the same latent ability nor was there any confirmatory effort on whether the estimated item parameters generalize to held-out models.

While the suggested methods seems interesting, it is not possible for me to interpret and evaluate the results from section 3 and 4.

**Requested Changes:**

**Assumptions**

The paper makes several, strong assumptions that need more justifications. For example, the normal distribution of the anchor ability $\theta_1$. This is a standard assumption for IRT in psychology but is, in my opinion not yet, established for LLMs. The $|\phi |<1 $ constraint is also not clear to me and needs more justification, at least some comments.

Constraining the discrimination parameters of the items to be always positive is what it should be, but requiring it seems to me like a major problem. If the model actually estimates zero or negative discrimmination of an item, that could hint to it being falsly labeled. It could also hint to a benchmark measuring multiple different latent abilities. Therefore, enforcing the slopes to be positive seems to me like a major disadvantage and the design needs more justification.

Assuming constant item parameters over time is reasonable for estimation reasons but also requires more justification. This is related to a required discussion of what the selected benchmarks measure and whether they measure the same latent variable.


**Quality & Fit**

A quality and model fit discussion should be introduced with two goals in mind:

1. establishing that the items within one of the selected benchmarks measure the same latent variable

2. establishing that the proposed model can better explain the variance in item responses, given the latent trait measured, as compared with existing methods (even if that is a purely accuracy-based analysis)

For 1) I suggest PCA or exploratory factor analysis, Since these benchmarks are not established measurement instruments, I suspect that many items should be removed before applying IRT. This should be done on a separate set of models.

For 2) I suggest common fit metrics like RMSEA, CFI, TLI and item residuals. I would compare this to standard Rasch, 2PL and 3PL models without the autoregressive components or other variants of the proposed model in order to establish the proposed choice. I would also test whether the estimated item parameters generalize to other heldout models. It would also be good to have a clear, quantifiable test or a prediction that shows why the proposed IRT method is better than purely accuracy-based analysis.

---

### Review · Reviewer_bP6Q · 2026-06-29

**Summary Of Contributions:**

The paper proposes a dynamic IRT framework for longitudinal LLM evaluation: a standard 2PL IRT model (item difficulty + discrimination, per-snapshot latent ability) augmented with an AR(1) prior on the ability sequence, fit jointly by MAP. It is applied to four multiple-choice benchmarks across 18 GPT snapshots (plus simple DeepSeek/Llama sequences in Appendix).

**Strengths**
- The idea of framing longitudinal LLM evaluation as temporal inference over a latent ability trajectory is interesting and novel to my knowledge.
- The paper is easy to follow.

**Weaknesses**
- No static-IRT baseline, so the dynamic model’s claimed advantage is not justified
- The AR(1) prior is misspecified for real world scenarios where models are usually released as a tree spanning different sizes instead of only a chronological sequence.
- Some fitted parameters are strangely large (e.g. discrimination values ~10³–10⁴in Table 5).

**Audience:**

Yes

**Audience Explanation:**

The idea of framing longitudinal LLM evaluation as temporal inference over a latent ability trajectory is interesting and novel to my knowledge.

**Broader Impact Concerns:**

No dedicated statement needed.

**Claims And Evidence:**

No

**Claims Explanation:**

The math is correct and descriptive results are clear, but the comparative claims are unsupported. The paper’s thesis is that dynamic IRT beats static IRT, yet runs no static baseline, ablation, simulation, or held-out metric. It shows the model can be fit, not that the AR(1) layer helps.

**Requested Changes:**

- Add a static-IRT baseline and quantify the dynamic model’s advantage on a held-out predictive metric.
- Report uncertainty on ability trajectories and on the Δp regression signals.
- Extend the framework or at least discuss how to handle release trees.

---

### Review · Reviewer_jys4 · 2026-07-16

**Summary Of Contributions:**

This paper introduces a dynamic IRT framework for evaluating LLM families over time: a 2PL IRT model in which item difficulty/discrimination are calibrated jointly across snapshots, while each snapshot's latent ability evolves under an AR(1) prior. Applied to various chronological GPT snapshots across four multiple-choice benchmarks, the main finding is that GPT-5.2 regresses relative to GPT-5 despite similar aggregate accuracy.

**Audience:**

Yes

**Audience Explanation:**

The motivating problem -- aggregate accuracy masking item-level heterogeneity and silent regressions across LLM releases -- is well-motivated and relevant, and applying longitudinal IRT here is a reasonable, underexplored direction. Readers in benchmark design and evaluation methodology would likely find this valuable, though the specific empirical claims need firmer statistical grounding.

**Broader Impact Concerns:**

No major concerns beyond what is typical for LLM benchmarking papers; no human subjects or sensitive data involved.

**Claims And Evidence:**

No

**Claims Explanation:**

The headline claim rests on point estimates alone -- no standard errors or confidence intervals are reported anywhere, despite the fact that larger samples are needed for stable estimates. The appendix item tables show discrimination estimates ranging from exactly 0.000 to values like 16241.319 -- several orders of magnitude beyond typical 2PL scales, with many items sharing identical (b, a) values. With only 18 (or 5-6) snapshots used to jointly fit two parameters per item, this pattern is consistent with unstable estimation, which undermines the item-level localization claim.

**Requested Changes:**

1. Report uncertainty (SEs/CIs) for ability and item parameter estimates, and propagate this into the GPT-5.2 vs. GPT-5 comparison.
2. Address the boundary-degenerate AR(1) fits and extreme/zero discrimination estimates — clarify whether these reflect true convergence, separation artifacts, or a need for regularization.